# Towards Understanding Extrapolation: a Causal Lens

**Lingjing Kong**[1][*]    **Guangyi Chen**[1,2][*]    **Petar Stojanov**[3]    **Haoxuan Li**[2]    **Eric P. Xing**[1,2]
**Kun Zhang**[1,2]

[1] Carnegie Mellon University
[2] Mohamed bin Zayed University of Artificial Intelligence
[3] Broad Institute of MIT and Harvard, Cancer Program, Eric and Wendy Schmidt Center

## Abstract

Canonical work handling distribution shifts typically necessitates an entire target distribution that lands inside the training distribution. However, practical scenarios often involve only a handful of target samples, potentially lying outside the training support, which requires the capability of extrapolation. In this work, we aim to provide a theoretical understanding of when extrapolation is possible and offer principled methods to achieve it without requiring an on-support target distribution. To this end, we formulate the extrapolation problem with a latent-variable model that embodies the minimal change principle in causal mechanisms. Under this formulation, we cast the extrapolation problem into a latent-variable identification problem. We provide realistic conditions on shift properties and the estimation objectives that lead to identification even when only one off-support target sample is available, tackling the most challenging scenarios. Our theory reveals the intricate interplay between the underlying manifold's smoothness and the shift properties. We showcase how our theoretical results inform the design of practical adaptation algorithms. Through experiments on both synthetic and real-world data, we validate our theoretical findings and their practical implications.

## 1 Introduction

Extrapolation necessitates the capability of generalizing beyond the training distribution support, which is essential for the robust deployment of machine learning models in real-world scenarios. Specifically, given access to a source distribution $\mathcal{D}_{\mathrm{src}} := p(\mathbf{x}_{\mathrm{src}}, \mathbf{y}_{\mathrm{src}})$ with support $\mathcal{X}_{\mathrm{src}} := \mathrm{Supp}(p_{\mathrm{src}}(\mathbf{x}))$ and one or a few out-of-support samples $\mathbf{x}_{\mathrm{tgt}} \notin \mathcal{X}_{\mathrm{src}}$, the goal of extrapolation is to predict the target label $\mathbf{y}_{\mathrm{tgt}}$. For example, if the training distribution includes dog images, we aim to accurately classify dogs under unseen camera angles, lighting conditions, and backgrounds. While intuitive for humans, machine learning models can be brittle to minor distribution shifts [1–4].

Addressing distribution shifts has garnered significant attention from the community. Unsupervised domain adaptation under covariate shifts addresses the shift of the marginal distribution $p(\mathbf{x})$ across domains. However, canonical techniques such as importance sampling and re-weighting [5–9] are predicated on the assumptions of overlapping supports $\mathrm{Supp}(p_{\mathrm{tgt}}(\mathbf{x})) \subset \mathrm{Supp}(p_{\mathrm{src}}(\mathbf{x}))$ and the availability of the entire target marginal distribution $p_{\mathrm{tgt}}(\mathbf{x})$. Similarly, domain generalization [10–12] assumes access to multiple source distributions $p_{\mathrm{src}}(\mathbf{x}, \mathbf{y})$ whose supports jointly cover the target distribution. In addition to these methods, test-time adaptation (TTA) [13–16] is particularly relevant to our discussion of extrapolation. TTA addresses out-of-distribution test samples at the individual sample level. Canonical methods include updating the source model with entropy-based or self-supervised losses on target samples. However, most TTA research focuses on empirical aspects, with limited theoretical formalization [17]. Most related to our work, Kong et al. [18] and Li et al.

---

[*] Equal Contribution.

38th Conference on Neural Information Processing Systems (NeurIPS 2024).

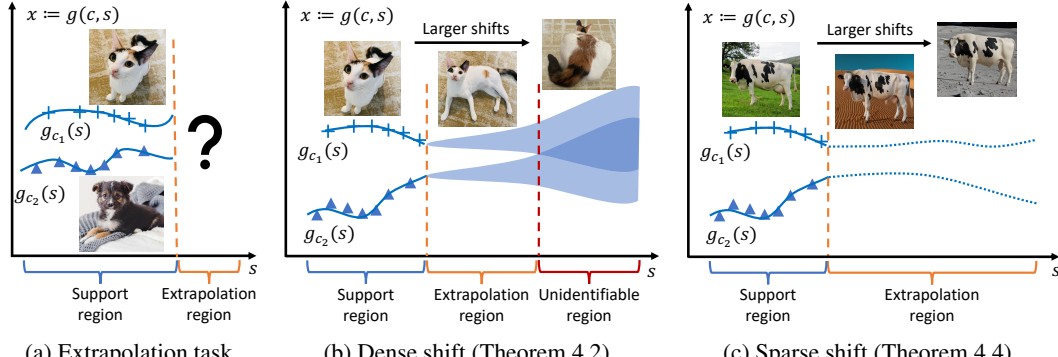

(a) Extrapolation task.  (b) Dense shift (Theorem 4.2).  (c) Sparse shift (Theorem 4.4).

Figure 1: **Illustration of extrapolation and our theoretical conditions.** The horizontal axis represents the changing variable $\mathbf{s}$, ranging from the source support to out-of-support regions. The vertical axis represents the observed data $\mathbf{x}$ living on the manifolds indexed by different values of the invariant variable $\mathbf{c}$. Figure (a) demonstrates that given a point out of support it is unclear which class manifolds it belongs to. Figure (b) illustrates the dense shift condition (Theorem 4.2) where $\mathbf{s}$ potentially changes all pixels in the images, such as the camera angle in the example. In this case, we can identify the invariant variable $\mathbf{c}$ under a moderate amount of shift until the shift becomes excessive. For instance, the back view of the cat in the figure could be confused with other animals. Figure (c) illustrates the sparse shift condition (Theorem 4.4) where $\mathbf{s}$ influences a limited number of pixels, such as the background in the example. In contrast to the dense shift, we can identify $\mathbf{c}$ under the sparse shift regardless of its severity. In the figure, there is no ambiguity of the class "cow" even though the background has changed to the moon.

[19] propose theoretical frameworks to characterize distribution shifts and explore conditions for identifying latent changing factors. However, these frameworks assume access to multiple source distributions with overlapping supports, which are not directly applicable to the extrapolation problem considered in this work, where we have potentially only one out-of-support target sample $\mathbf{x}_{\text{tgt}}$.

Since the target $\mathbf{y}_{\text{tgt}}$ can be arbitrarily out of the source support (Figure 1a), extrapolation is ill-posed without proper assumptions on the relationship between the source and the target. In this work, we formulate a latent-variable model that encodes a *minimal change principle* to address this ill-posedness. Specifically, we assume that a latent variable $\mathbf{z}$ determines $\mathbf{x}$ such that $\mathbf{x} := g(\mathbf{z})$. The minimal change principle entails the following two assumptions on the generating process. 1) The out-of-support nature of $\mathbf{x}_{\text{tgt}}$ stems from only a *subspace* of $\mathbf{z}$, denoted as $\mathbf{s}$, while the complement partition $\mathbf{c}$ for the target sample $\mathbf{x}_{\text{tgt}}$ is within the training support, i.e., $\mathbf{z} := [\mathbf{c}, \mathbf{s}]$, $\mathbf{s}_{\text{tgt}} \notin \mathcal{S}_{\text{src}}$, and $\mathbf{c}_{\text{tgt}} \in \mathcal{C}_{\text{src}}$. 2) The changing variable $\mathbf{s}$ only controls non-semantic attributes in $\mathbf{x}$ and thus doesn't influence the label $\mathbf{y}$, i.e., $\mathbf{y} := g_y(\mathbf{c})$. This formulation attributes the seemingly complex shifts in the pixel space of $\mathbf{x}$ to simple intrinsic changes (in the sense that this change involves only $\mathbf{s}$) in the latent space, allowing us to reason about the transfer via the invariant variable $\mathbf{c}$. Under our formulation, extrapolation amounts to identifying invariant latent variables $\mathbf{c}$, with which a model $f : \mathbf{c} \mapsto \mathbf{y}$ trained on the labeled source dataset can be directly applied to the target sample.

In light of this formulation, we investigate the identifiability conditions of the invariant variable $\mathbf{c}$. We propose two sets of conditions addressing two regimes of the influence from $\mathbf{s}$ on $\mathbf{x}$. We refer to the case where all dimensions $\mathbf{x}_i$ (e.g., all pixels in an image) could be influenced by the changing variable $\mathbf{s}$ as the dense influence and the case where only a limited subset of dimensions $x_i$ is affected as the sparse influence. Specifically, our first condition (Dense Influence, Theorem 4.2) states that if $\mathbf{s}$'s influence is dense, then assuming that $\mathbf{c}$ takes on only finite values, extrapolation requires the manifold associated with each value of $\mathbf{c}$ (i.e., $g(\mathbf{c}, \cdot)$ over $\mathbf{s}$) to be adequately separable, and the target changing variable $\mathbf{s}_{\text{tgt}}$ to be close to the source support $\mathcal{S}_{\text{src}}$. Intuitively, if images from two classes (two values of $\mathbf{c}$) are sufficiently distinguishable, such as cats and dogs, we can still recognize the class of the target sample $\mathbf{x}_{\text{tgt}}$, even if it has undergone moderate unseen shifts on all pixels like camera angles and positions controlled by $\mathbf{s}$ (Figure 1b). Our second condition (Sparse Influence, Theorem 4.4) states that if $\mathbf{s}$'s influence is sufficiently sparse, then extrapolation can occur regardless of the distance of $\mathbf{s}_{\text{tgt}}$ to the source support $\mathcal{S}_{\text{src}}$. Intuitively, we can recognize the class "cow" even if only the background is changed, regardless of the severity (Figure 1c).

Our theory provides insights into the interaction between the underlying manifold's smoothness, the out-of-support distance, and the nature of the shift. We conduct synthetic data experiments to validate our theoretical results. Moreover, we discuss the relationship between our results and TTA approaches. In particular, we apply our theoretical insights to improve autoencoder-based MAE-TTT [20] and observe noticeable improvements. Additionally, we demonstrate that basic principles (sparsity constraints) from our framework can benefit the state-of-the-art TTA approach TeSLA [21]. Our empirical results not only show the practical viability of our theory but also pave the way for further advancements in the field.

In summary, our contributions are threefold:

- We formulate the extrapolation task as a latent-variable identification problem. Our latent-variable model encodes complex changes in observed variables to a partition of latent variables, allowing us to reason about transferability from the source to the target through latent variable identification.

- We provide identification guarantees for the proposed latent-variable model, including shifts of distinct properties (dense vs. sparse) and corresponding conditions on the generating process. Our theory provides an essential understanding of when latent-variable identification is possible without accessing an entire target distribution and assuming overlapping supports as in prior work [18, 19].

- Inspired by our theory, we propose to add a likelihood maximization term to autoencoder-based MAE-TTT [20] to facilitate the alignment between the target sample and the source distribution. In addition, we propose sparsity constraints to enhance the state-of-the-art TTA approach TeSLA [21]. We validate our proposals with empirical evidence.

## 2 Related Work

**Extrapolation.** Out-of-distribution generalization has attracted significant attention in recent years. Unlike our work, the bulk of the work is devoted to generalizing to target distributions on the same support as the source distribution [22, 23, 8]. Recent work [24–27] investigates extrapolation in the form of compositional generalization by resorting to structured generating functions (e.g., additive, slot-wise). Another line of work [28–30] studies extrapolation in regression problems and does not consider the latent representation. Saengkyongam et al. [31] leverage a latent variable model and linear relations between the interventional variable and the latent variable to handle extrapolation. In this work, we formulate extrapolation as a latent variable identification problem. Unlike the semi-parametric conditions in prior work, our conditions do not constrain the form of the generating function and are more compatible with deep learning models and tasks.

**Latent-variable identification for transfer learning.** In the latent-variable model literature, one often assumes latent variables $\mathbf{z}$ generate the observed data $\mathbf{x}$ (e.g., images, text) through a generating function. However, the nonlinearity of deep learning models requires the generating function to be nonlinear, which has posed major technical difficulty in recovering the original latent variable [32]. To overcome this setback, a line of work [33–36] assumes the availability of an auxiliary label $\mathbf{u}$ for each sample $\mathbf{x}$ and under different $\mathbf{u}$ values, each component $z_i$ of $\mathbf{z}$ experiences sufficiently large shift in its distribution. Since this framework assumes all latent components' distributions vary over distributions indexed by $\mathbf{u}$, it does not assume the existence of some shared, invariant information across distributions, which is often the case for transfer learning tasks. To address this issue, recent work [18, 19] introduce a partition of $\mathbf{z}$ into an invariable variable $\mathbf{c}$ and an changing variable $\mathbf{s}$ (i.e., $\mathbf{z} := [\mathbf{c}, \mathbf{s}]$) such that $\mathbf{c}$'s distribution remains constant over distributions. They show both $\mathbf{c}$ and $\mathbf{s}$ can be identified and one can directly utilize the invariant variable $\mathbf{c}$ for domain adaptation. However, their techniques crucially rely on the variability of the changing variable $\mathbf{s}$, mandating the availability of multiple sufficiently disparate distributions (including the target) and their overlapping supports. These constraints make them unsuitable for the extrapolation problem. In comparison, our theoretical results give identification of the invariant variable $\mathbf{c}$ (the on-support variable in the extrapolation context) with only one source distribution $p_{\text{src}}(\mathbf{x})$ and as few as one out-off support target sample $\mathbf{x}_{\text{tgt}}$ through mild assumptions on the generating function, directly tackling the extrapolation problem.

Please refer to Section A1 for more related work.

# 3 Extrapolation and Latent-Variable Identification

Given the labeled source distribution $p(\mathbf{x}_{\mathrm{src}}, \mathbf{y}_{\mathrm{src}})$, our goal is to make predictions on a target sample $\mathbf{x}_{\mathrm{tgt}}$ outside the source support ($\mathbf{x}_{\mathrm{tgt}} \notin \mathcal{X}_{\mathrm{src}}$). While more target samples would provide better information about the distribution shift, in practice, we often have only a handful of samples to work with. Therefore, we focus on the challenging scenario where only one target sample $\mathbf{x}_{\mathrm{tgt}}$ is available.

Making reliable predictions on out-of-support samples $\mathbf{x}_{\mathrm{tgt}}$ is infeasible without additional structure. Real-world problems where humans successfully extrapolate often follow a minimal change principle: they involve sparse, non-semantic intrinsic shifts despite complex raw data changes. For example, a person who has only seen a cow on a pasture can recognize the same cow on a beach, even if the background pixels change significantly. Here, the cow corresponds to the part of the latent variable that remains within the support of the source data, which we call the invariant variable $\mathbf{c}$ ($\mathbf{c}_{\mathrm{tgt}} \in \mathcal{C}_{\mathrm{src}}$), while the background change corresponds to the complement that drifts off the source support, which we call the changing variable $\mathbf{s}$ ($\mathbf{s}_{\mathrm{tgt}} \notin \mathcal{S}_{\mathrm{src}}$). Clearly, extrapolation is impossible if the intrinsic shift is dense (i.e., all dimensions change, $\mathbf{z} = \mathbf{s}$) or semantic (i.e., $\mathbf{y}$ is a function of $\mathbf{s}$). For instance, if the variable $\mathbf{s}$ also alters the cow's appearance drastically, making it unrecognizable, extrapolation fails. We define the data-generating process to encapsulate this minimal change principle, as follows:

$$\mathbf{c} \sim p(\mathbf{c}), \ \mathbf{s} \sim p(\mathbf{s}|\mathbf{c});$$
$$\mathbf{x} = g(\mathbf{z}), \ \mathbf{y} = g_{\mathbf{y}}(\mathbf{c}). \tag{1}$$

Figure 2: **The data-generating process.** The invariant latent variable $c$ and the changing latent variable $s$ jointly generate the observed variable $x$. The dashed line indicates potential statistical dependence.

In this process, the latent space $\mathbf{z} \in \mathcal{Z} \subset \mathbb{R}^{d_\mathbf{z}}$ comprises two subspaces: the invariant variable $\mathbf{c} \in \mathcal{C} \subset \mathbb{R}^{d_\mathbf{c}}$ and the changing variable $\mathbf{s} \in \mathcal{S} \subset \mathbb{R}^{d_\mathbf{s}}$. We define $\mathcal{Z} := \mathcal{Z}_{\mathrm{src}} \cup \{\mathbf{z}_{\mathrm{tgt}}\}$ as the source support augmented with the target sample $\mathbf{z}_{\mathrm{tgt}}$ and similarly $\mathcal{X}$ and $\mathcal{S}$. The invariant variable $\mathbf{c}$ encodes shared information between the source distribution $p(\mathbf{x}_{\mathrm{src}})$ and the out-of-support target sample $\mathbf{x}_{\mathrm{tgt}}$, while the changing variable $\mathbf{s}$ describes the shift from the source support $\mathcal{X}_{\mathrm{src}}$. Hence, $\mathbf{c}_{\mathrm{tgt}} \in \mathcal{C}_{\mathrm{src}}$ and $\mathbf{s}_{\mathrm{tgt}} \notin \mathcal{S}_{\mathrm{src}}$. The variables $\mathbf{z} := [\mathbf{c}, \mathbf{s}]$ jointly generate the observed variable $\mathbf{x} \in \mathcal{X} \subset \mathbb{R}^{d_\mathbf{x}}$ through an invertible generating function $g : \mathbb{R}^{d_\mathbf{z}} \to \mathbb{R}^{d_\mathbf{x}}$. Furthermore, we assume that the label $\mathbf{y}$ originates from the invariant variable $\mathbf{c}$. This assumption reflects the reality that factors such as camera angles and lighting do not affect the object's class in an image.

Our latent-variable model adheres to the minimal change principle in two key ways: (1) the target sample's out-of-support nature arises from only a subset of latent variables $\mathbf{s}$, and (2) these changing variables $\mathbf{s}$ are non-semantic, thus not influencing the label $\mathbf{y}$.

**Extrapolation and identifiability.** Under this framework, extrapolation is possible if we can identify the true invariant variable $\mathbf{c}$ in both the source distribution $p_{\mathrm{src}}(\mathbf{x})$ and the target data $\mathbf{x}_{\mathrm{tgt}}$. This allows us to learn a classifier $f_{\mathrm{cls}} : \mathbf{c} \mapsto \mathbf{y}$ on the labeled source distribution $p_{\mathrm{src}}(\mathbf{x}, \mathbf{y})$. Since the target sample's invariant variable falls within the source support ($\mathbf{c}_{\mathrm{tgt}} \in \mathcal{C}_{\mathrm{src}}$), this classifier $f_{\mathrm{cls}}$ can be directly applied to the target sample $\mathbf{c}_{\mathrm{tgt}}$. Thus, the task of extrapolation reduces to identifying the invariant variable $\mathbf{c}$ in both the source distribution $p(\mathbf{x}_{\mathrm{src}})$ and the target sample $\mathbf{x}_{\mathrm{tgt}}$. In Section 4, we explore the conditions for identifying the invariant variable $\mathbf{c}$.

Given the above reasoning, we define identifiability in Definition 3.1 (i.e., block-wise identifiability [37, 24]) which suffices for extrapolation.

**Definition 3.1** (Identifiability of the Invariant Variable $\mathbf{c}$)**.** For any $\mathbf{x}_1$ and $\mathbf{x}_2$, their true invariant variables $\mathbf{c}_1, \mathbf{c}_2$ are equal if and only if the estimates $\hat{\mathbf{c}}_1, \hat{\mathbf{c}}_2$ are equal: $\mathbf{c}_1 = \mathbf{c}_2 \iff \hat{\mathbf{c}}_1 = \hat{\mathbf{c}}_2$.

# 4 Identification Guarantees for Extrapolation

In this section, we provide two sets of conditions on which one can identify the invariant variable $\mathbf{c}$ and discuss the intuition and implications.

As discussed in Section 3, we need to identify the target sample $\mathbf{x}_{\mathrm{tgt}}$ with source samples $\mathbf{x}_{\mathrm{src}}$ that share the same invariant variable values with the target sample, i.e., $\mathbf{c}_{\mathrm{src}} = \mathbf{c}_{\mathrm{tgt}}$. This enables us to obtain the label of $\mathbf{x}_{\mathrm{tgt}}$ by assigning the label of such $\mathbf{x}_{\mathrm{src}}$. The shift between the source distribution

$p(\mathbf{x}_{\mathrm{src}})$ and the target sample $\mathbf{x}_{\mathrm{tgt}}$ originates from the out-of-support nature of the changing variable $\mathbf{s}_{\mathrm{tgt}}$, i.e., $\mathbf{s}_{\mathrm{tgt}} \notin \mathcal{S}_{\mathrm{src}}$, it is crucial to impose proper assumptions on the influence of $\mathbf{s}$ on $\mathbf{x}$ so that $\mathbf{x}$ retains sufficient footprints of $\mathbf{c}$ for identification beyond the source support.

We denote the Jacobian matrix of the generating function $g$ as $\mathbf{J}_g(\mathbf{z})$ and $\mathbf{x}$'s dimensions under the influence of $\mathbf{s}$ as $\mathcal{I}_{\mathbf{s}}(\mathbf{z}) := \{i \in [d_{\mathbf{x}}] : \exists j \in \{d_{\mathbf{c}} + 1, \ldots, d_{\mathbf{z}}\}, \text{s.t.}, [\mathbf{J}_g(\mathbf{z})]_{i,j} \neq 0\}$. We note that the set $\mathcal{I}_{\mathbf{s}}(\mathbf{z})$ is a function of $\mathbf{z}$, since the influenced dimensions may vary over $\mathbf{z}$. Intuitively, if $\mathbf{s}$ influences $\mathbf{x}$ in a dense manner, i.e., large $|\mathcal{I}_{\mathbf{s}}(\mathbf{z})|$ for potentially all dimensions $x_i$, there may not be dimensions of $\mathbf{x}$ serving as clear signatures of $\mathbf{c}$, thereby increasing the difficulty of identify $\mathbf{c}$. Additionally, the degree to which the changing variable $\mathbf{s}$ is out-of-support plays a critical role – the further the target changing variable– the further the target changing variable $\mathbf{s}_{\mathrm{tgt}}$ deviates from the source distribution support $\mathcal{S}_{\mathrm{src}}$, the more severe and unpredictable the shift becomes, making it harder to retrieve $\mathbf{c}$. In the following, we formalize conditions on the influence of $\mathbf{s}$ from these two perspectives, revealing an interesting trade-off and interaction between these factors.

**Notations.** The true generating process involves $\mathbf{c}$, $\mathbf{s}$, distributions $p$, and $g$ (Equation 1), we define their statistical estimates with $\hat{\mathbf{c}}$, $\hat{\mathbf{s}}$, $\hat{p}$, and $\hat{g}$ through the objectives we will introduce.[2] We assume that the estimation process respects the conditions of the corresponding true-generating process.

### 4.1 Dense-shift Conditions

We begin by investigating scenarios where there are no constraints on the number of dimensions of $\mathbf{x}$ (i.e., the number of pixels) influenced by the changing variable $\mathbf{s}$, i.e., potentially large $|\mathcal{I}_{\mathbf{s}}(\mathbf{z})|$, which we term as dense shifts. For images, these shifts encompass global transformations such as changes in camera angles and lighting conditions that could potentially affect all pixel values (Figure 1b).

**Understanding the problem.** As dense shifts could influence all the dimensions of $\mathbf{x}$, every dimension could be out of the source support and there might not be dimensions of $\mathbf{x}$ that solely contain the information of $\mathbf{c}$. Consequently, relying on any subset of $\mathbf{x}$ dimensions to infer the original $\mathbf{c}$ becomes untenable. For instance, consider a scenario where the source distribution contains frontal-view images of a cat, while the target sample portrays the same cat from a side view (Figure 1b). The model cannot recognize these two images as the same cat (the same $\mathbf{c}$) by matching a specific part of the side view, say the cat's nose, to samples in the source distribution because this cat's nose only shows up as a front view and can be vastly different in terms of the pixel region and values. The model cannot match specific features such as the cat's nose, between the side-view target and the source distribution, as the pixel region and values for the nose drastically differ.

**Our approach.** For the reasons above, we need to constrain such dense changes so that even when all dimensions are affected, the target sample adheres to some intrinsic structure determined by the underlying $\mathbf{c}_{\mathrm{tgt}}$ and remains distinguishable from samples of $\mathbf{c} \neq \mathbf{c}_{\mathrm{tgt}}$ In many real-world distributions, we can interpret $\mathbf{c}$ as the embedding vector of classes or other categories, with each $\mathbf{c}$ value indexing a manifold $g(\mathbf{c}, \cdot)$ over $\mathbf{s}$. If manifolds are smooth and sufficiently separable from each other, they should exhibit limited variations in the adjacent region to the training support, avoiding confusion between distinct categories. For example, there exists a noticeable distinction between cats and lions, such that moderate illumination changes would not cause confusion until illumination significantly obscures distinguishing features. In the following, we formalize these structures by assuming a finite cardinality of $\mathbf{c}$ and constraining the distance of $\mathbf{s}_{\mathrm{tgt}}$ to the support $\mathcal{S}_{\mathrm{src}}$.

**Additional notations.** We denote with $J_u$ an upper bound of the Jacobian spectrum norm: $\|\mathbf{J}_g(\mathbf{z})\| \leq J_u$ on the support. In Appendix A2, we show $J_u < \infty$ due to Assumption 4.1-i and Assumption 4.1-ii. We denote with $D(\mathbf{c}_1, \mathbf{c}_2)$ the $\ell_2$ distance between two manifolds on the support boundary: $D(\mathbf{c}_1, \mathbf{c}_2) := \inf_{\mathbf{s}_1, \mathbf{s}_2 \in \mathrm{Bd}(\mathcal{S}_{\mathrm{src}})} \|g(\mathbf{c}_1, \mathbf{s}_1) - g(\mathbf{c}_2, \mathbf{s}_2)\|$, where we denote the boundary of source support with $\mathrm{Bd}(\mathcal{S}_{\mathrm{src}})$. We denote with $D(\mathbf{s}, \mathcal{S}_{\mathrm{src}})$ the minimal $\ell_2$ distance between $\mathbf{s}$ and the source support $\mathcal{S}_{\mathrm{src}}$, i.e., $D(\mathbf{s}, \mathcal{S}_{\mathrm{src}}) := \inf_{\mathbf{s}_{\mathrm{src}} \in \mathcal{S}_{\mathrm{src}}} \|\mathbf{s} - \mathbf{s}_{\mathrm{src}}\|$.

**Assumption 4.1** (Identification Conditions under Global Shifts)**.**

   i *[Smoothness & Invertibility]: The generating function g in Equation 1 is a smooth invertible function with a smooth inverse everywhere.*

   ii *[Compactness]: The source data space $\mathcal{X}_{\mathrm{src}} \subset \mathbb{R}^{d_x}$ is closed and bounded.*

---

[2]We slightly abuse the notation $p$ to denote density functions for continuous variables or delta functions for discrete variables.

*iii [Discreteness]: The invariant variable $\mathbf{c}$ takes on values from a finite set: $\mathcal{C} = \{\mathbf{c}_k\}_{k \in [K]}$.*

*iv [Continuity]: The probability density function $p(\mathbf{s}|\mathbf{c})$ is continuous over $\mathbf{s} \in \mathcal{S}_{\mathrm{src}}$, for all $\mathbf{c} \in \mathcal{C}$.*

*v [Out-of-support Distance]: The target sample's out-support components $\mathbf{s}_{\mathrm{tgt}}$'s distance from the source support $\mathcal{S}_{src}$ is constrained: $\inf_{\mathbf{s} \in \mathcal{S}_{src}} \|\mathbf{s}_{\mathrm{tgt}} - \mathbf{s}\| \leq \frac{\min_{\mathbf{c} \in \mathcal{C} \setminus \{\mathbf{c}_{\mathrm{tgt}}\}} D(\mathbf{c}_{\mathrm{tgt}}, \mathbf{c})}{2 J_u}$.*

**Discussion on the conditions.** As discussed above, the main conditions revolve two key factors: the discrete structure of the invariant distribution of $p(\mathbf{c})$ in Assumption 4.1-iii and the off-support distance of the changing variable $\mathbf{s}$ in Assumption 4.1-v. The discrete structure of $p(\mathbf{c})$ is applicable to many real-world scenarios, especially classification tasks where the semantic invariant information often manifests as discrete class labels or other categorical distinctions. While this assumption is typically valid, it can be extended to encompass continuous dimensions in the invariant variable $\mathbf{c} := [\mathbf{c}_c, \mathbf{c}_d]$ where $\mathbf{c}_c$ and $\mathbf{c}_d$ stand for the continuous and discrete dimensions, respectively. In such cases, we can group the continuous dimensions $\mathbf{c}_c$ with the changing variable $\mathbf{s}$ and the same proof would give rise to the identification of the discrete part $\mathbf{c}_d$, which suffices for classification tasks. The off-support distance condition involves the smoothness of the generating function $g$, where a smoother generating function allows more leeway for the target changing variable $\mathbf{s}_{\mathrm{tgt}}$ to deviate. When $\mathbf{s}$ controls the camera angle, one may be able to recognize a slightly sided view of cats after seeing front views in the source until the $\mathbf{s}$ deviates too far and all images become back views, potentially leading to confusion with other animals (Figure 1b). Assumption 4.1-i ensures that the generating process preserves the latent information, which is widely adopted in the literature [18, 19, 35, 36, 33, 38]. Specifically, this guarantees that manifolds indexed by distinct values of $\mathbf{c}$ are separate from each other, maintaining strictly positive distances between them. Assumption 4.1-ii,iv are technical conditions mirroring realities that pixels values are bounded and the changing variable $\mathbf{s}$ often represent attributes that vary gradually across its support (e.g., lighting and angles).

**Theorem 4.2** (Extrapolation under Dense Shifts). *Assuming a generating process in Equation 1, we estimate the distribution with model $(\hat{g}, \hat{p}(\hat{\mathbf{c}}), \hat{p}(\hat{\mathbf{s}}))$ with the objective:*

$$\sup \hat{p}(\hat{\mathbf{c}}_{\mathrm{tgt}}), \quad \textit{Subject to: } \hat{p}(\mathbf{x}) = p(\mathbf{x}), \, \forall \mathbf{x} \in \mathcal{X}_{\mathrm{src}}; \quad \hat{\mathbf{s}}_{\mathrm{tgt}} \in \arg \inf_{\hat{\mathbf{s}}} D(\hat{\mathbf{s}}, \hat{\mathcal{S}}_{\mathrm{src}}). \quad (2)$$

*Under Assumption 4.1, the estimated model can attain the identifiability in Definition 3.1.*

**Proof sketch.** We estimate the generative process through maximum likelihood estimation on the source distribution $\hat{p}(\hat{\mathbf{x}}) = p(\mathbf{x})$. Under Assumption 4.1-i,ii,iii,iv, we can establish the identification of $\mathbf{c}$ on the *support* [38], i.e., $\hat{\mathbf{c}}_1 = \hat{\mathbf{c}}_2 \iff \mathbf{c}_1 = \mathbf{c}_2$, for $\mathbf{x}_1, \mathbf{x}_2 \in \mathcal{X}_{\mathrm{src}}$. This implies that all samples $\mathbf{x}_{\mathrm{src}}$ on a given source manifold $g(\mathbf{c}, \cdot)$ share identical values of $\hat{\mathbf{c}}$. In the objective, we maximize the likelihood $\hat{p}(\hat{\mathbf{c}}_{\mathrm{tgt}})$ to drive $\hat{\mathbf{c}}_{\mathrm{tgt}}$ to match one of the discrete values of $\hat{\mathbf{c}}$ with nontrivial probability mass. Given the identification of $\mathbf{c}$ on the support, this equates to assigning $\mathbf{x}_{\mathrm{tgt}}$ to a manifold $g(\mathbf{c}^*, \cdot)$ with $\mathbf{c}^* \in \mathcal{C}$. Our task now switches to ensuring that this is the correct manifold for $\mathbf{x}_{\mathrm{tgt}}$, i.e., $\mathbf{c}^* = \mathbf{c}_{\mathrm{tgt}}$. To accomplish this, we select the estimated model that uses the minimal off-support distance on $\hat{\mathbf{s}}$ (i.e., $D(\hat{\mathbf{s}}, \hat{\mathcal{S}}_{\mathrm{src}})$) to explain the off-support nature of $\mathbf{x}_{\mathrm{tgt}}$. This also embodies the minimal change principle. This and Assumption 4.1-v guarantee that only the correct manifold ($g(\mathbf{c}_{\mathrm{tgt}}, \cdot)$) can effective capture $\mathbf{x}_{\mathrm{tgt}}$, thereby facilitating the desired identification. In practice, these constraints can be implemented through Lagrange multipliers. Full proof is given in Appendix A2.

### 4.2 Sparse-shift Conditions

We now examine cases where the changing variable $\mathbf{s}$ influences only a subset of dimensions of $\mathbf{x}$, i.e., a limited $|\mathcal{I}_{\mathbf{s}}(\mathbf{z})|$, which we refer to as sparse shifts. For image distributions, these shifts include local corruptions or background changes that do not alter foreground objects (Figure 1c).

**Additional notations.** We define the index set $\mathcal{I}_{\mathbf{c}}(\mathbf{z})$ under the influence of $\mathbf{c}$ and the indices under the the exclusive influence of $\mathbf{c}$ as $\mathcal{I}_{\mathbf{c} \setminus \mathbf{s}}(\mathbf{z}) := \mathcal{I}_{\mathbf{c}}(\mathbf{z}) \setminus \mathcal{I}_{\mathbf{s}}(\mathbf{z})$.

**Understanding the problem.** In contrast to the dense-shift scenario, here we have a non-trivial subset of dimensions $[\mathbf{x}]_{\mathcal{I}_{\mathbf{c} \setminus \mathbf{s}}(\mathbf{z})}$ that are unaffected by the changing variable $\mathbf{s}$. Consequently, if these dimensions carry sufficient information about $\mathbf{c}$, we can exploit them to directly recover the true $\mathbf{c}$, regardless of the distance $[\mathbf{x}]_{\mathcal{I}_{\mathbf{s}}(\mathbf{z})}$ deviates from the support. In contrast, in the dense-shift scenario, we need to constrain the out-of-support distance of $\mathbf{s}$ and assume the discreteness of $\mathbf{c}$. Consider

a scenario where a fixed $\mathbf{c}$ represents a specific cow and $\mathbf{s}$ controls only the background. Despite the variation in the target background (e.g., desert or space), we can effectively match the cow in the target image to the correct source images (see Figure 1c). While this may seem intuitive for humans, it is nontrivial for machine learning models to automatically recognize the region $[\mathbf{x}]_{\mathcal{I}_{\mathbf{c}\backslash\mathbf{s}}(\mathbf{z})}$, especially given its potential variation across $\mathbf{z}$.

**Our approach.** For image classification, $[\mathbf{x}]_{\mathcal{I}_{\mathbf{c}\backslash\mathbf{s}}(\mathbf{z})}$ corresponds to foreground objects (or a portion) unaffected under sparse changes induced by $\mathbf{s}$ (e.g., background changes). Humans can recognize this region because the pixels within it are strongly correlated (e.g., cow features). This observation motivates us to formalize such dependence structures in natural data to enable automatic identification.

**Assumption 4.3** (Identification Conditions under Local Shifts)**.**

  i *[Smoothness & Invertibility]: The generating function $g$ in Equation 1 is invertible and differentiable, and its inverse is also differentiable.*

  ii *[Invariant Variable Informativeness]: The dimensions under $\mathbf{c}$'s exclusive influence is uniquely determined: for a fixed $\mathbf{c} \in \mathcal{C}$, $[\mathbf{x}]_{\mathcal{I}_{\mathbf{c}\backslash\mathbf{s}}(\mathbf{c},\mathbf{s}_1)} \neq [\mathbf{x}]_{\mathcal{I}_{\mathbf{c}\backslash\mathbf{s}}(\mathbf{c}^*,\mathbf{s}_2)}$ for any $\mathbf{c}^* \neq \mathbf{c}$, $\mathbf{s}_1 \in \mathcal{S}$, and $\mathbf{s}_2 \in \mathcal{S}$.*

  iii *[Sparse Influence]: At any $\mathbf{z} \in \mathcal{Z}$, the changing variable $\mathbf{s}$ influences at most $d_{\mathbf{s}}$ dimensions of $\mathbf{x}$, i.e., $|\mathcal{I}_{\mathbf{s}}(\mathbf{z})| \leq d_{\mathbf{s}}$. Alternatively, the two variables $\mathbf{c}$ and $\mathbf{s}$ do not intersect on their influenced dimensions $\mathcal{I}_{\mathbf{c}}(\mathbf{z}) \cap \mathcal{I}_{\mathbf{s}}(\mathbf{z}) = \emptyset$.*

  iv *[Mechanistic Dependence]: For all $\mathbf{z}$, any nontrivial partition $\mathcal{P}_1, \mathcal{P}_2$ of the dimensions $\mathcal{I}_{\mathbf{c}\backslash\mathbf{s}}(\mathbf{z})$ yields dependence between the sub-matrices of the Jacobian $\mathbf{J}_g(\mathbf{z})$: $rank([\mathbf{J}_g(\mathbf{z})]_{\mathcal{I}_{\mathbf{c}\backslash\mathbf{s}}}(\mathbf{z})) < rank([\mathbf{J}_g(\mathbf{z})]_{\mathcal{P}_1}(\mathbf{z})) + rank([\mathbf{J}_g(\mathbf{z})]_{\mathcal{P}_2}(\mathbf{z}))$.*

**Discussion on the conditions.** Assumption 4.3-iii stipulates that the influence of $\mathbf{s}$ is sparse, either in terms of dimension counts or in its intersection with the influence from $\mathbf{c}$. It is noteworthy that while the influence is sparse, its location can vary over images, as indicated by the dependence of $\mathcal{I}_{\mathbf{s}}$ on $\mathbf{z}$. Consequently, it can capture diverse image corruptions and background changes. Assumption 4.3-ii ensures that $[\mathbf{x}]_{\mathcal{I}_{\mathbf{c}\backslash\mathbf{s}}}$ is sufficiently informative about $\mathbf{c}$. For instance, it precludes scenarios where a sparse corruption alters the top stroke of "7" to resemble "1", rendering the uncorrupted region fundamentally unidentifiable. Assumption 4.3-iv enforces the dependence alluded to in our previous discussion: the unaffected dimensions $[\mathbf{x}]_{\mathcal{I}_{\mathbf{c}\backslash\mathbf{s}}}$ exhibit mechanistic dependence across them, characterized by the Jacobian rank [39]. Thus, generating separate parts of an object necessitates more capacity than generating the entire object, as the dependence across the two parts can inform each other's generation. This inherent dependence enables the identification of the unaffected region.

**Theorem 4.4** (Extrapolation under Sparse Shifts)**.** *Assuming a generating process in Equation 1, we estimate the distribution with model $(\hat{g}, \hat{p}(\hat{\mathbf{c}}), \hat{p}(\hat{\mathbf{s}}))$ with the objective:*

$$\sup \hat{p}(\hat{\mathbf{c}}_{\text{tgt}}), \quad \textit{Subject to:} \quad \hat{p}(\mathbf{x}) = p(\mathbf{x}), \forall \mathbf{x} \in \mathcal{X}_{\text{src}}. \tag{3}$$

*Under Assumption 4.3, the estimated model can attain the identifiability in Definition 3.1.*

**Proof sketch.** Maximizing the likelihood $\hat{p}(\hat{\mathbf{c}}_{\text{tgt}})$ assigns a value $\hat{\mathbf{c}}^* \in \hat{\mathcal{C}}$ to $\hat{\mathbf{c}}_{\text{tgt}}$. Building on our motivation, we leverage mechanistic dependence (Assumption 4.3-iv) to identify the unaffected dimension indices $\mathcal{I}_{\mathbf{c}\backslash\mathbf{s}}(\mathbf{z}) \subset [d_{\mathbf{x}}]$ with our estimated model. In other words, we have $\mathcal{I}_{\mathbf{c}\backslash\mathbf{s}}(\mathbf{z}) = \hat{\mathcal{I}}_{\mathbf{c}\backslash\mathbf{s}}(\hat{\mathbf{z}})$. Consequently, the unaffected dimensions in the estimated variable equal their counterparts in the true model: $[\mathbf{x}_{\text{tgt}}]_{\hat{\mathcal{I}}_{\mathbf{c}\backslash\mathbf{s}}(\hat{\mathbf{c}}^*,\hat{\mathbf{s}}^*)} = [\mathbf{x}_{\text{tgt}}]_{\mathcal{I}_{\mathbf{c}\backslash\mathbf{s}}(\mathbf{c}_{\text{tgt}},\mathbf{s}_{\text{tgt}})}$. Furthermore, Assumption 4.3-ii stipulates that the dimensions in the target sample $[\mathbf{x}_{\text{tgt}}]_{\mathcal{I}_{\mathbf{c}\backslash\mathbf{s}}(\mathbf{c}_{\text{tgt}},\mathbf{s}_{\text{tgt}})}$ cannot be attained by other $\mathbf{c} \neq \mathbf{c}_{\text{tgt}}$, so we have established that $\hat{\mathbf{c}}^*$ corresponds to the correct value $\mathbf{c}_{\text{tgt}}$. Full proof is in Appendix A3.

It's worth noting that unlike the global shift case (Theorem 4.2), here we do not place a constraint on the out-of-support-ness of $\mathbf{s}_{\text{tgt}}$, a point we empirically verify in Section 6.3.

### 4.3  Implications for Practical Algorithms

**Generative adaptation.** Our theoretical framework, inherently a generative model, can be implemented through auto-encoding over the source distribution and the target. Akin to our estimation framework, MAE-TTT [20] trains a masked auto-encoding model ($f_{\text{enc}}$ and $f_{\text{dec}}$) on the source distribution and adapts to target samples through the auto-encoding objective. Consequently, we have

Table 1: **Synthetic data test accuracy** under both dense and sparse shifts across a range of distances.

| Shifts | Dense | | | | Sparse | | | |
|---|---|---|---|---|---|---|---|---|
| Distance | 12.0 | 18.0 | 24.0 | 30.0 | 18.0 | 24.0 | 30.0 | 36.0 |
| Only Source | 0.59 | 0.55 | 0.45 | 0.45 | 0.54 | 0.54 | 0.56 | 0.52 |
| iMSDA [18] | 0.46 | 0.48 | 0.48 | 0.50 | 0.50 | 0.36 | 0.40 | 0.54 |
| **Ours** | **0.78** | **0.69** | **0.72** | **0.72** | **0.72** | **0.72** | **0.76** | **0.70** |

$f_{\text{dec}}(f_{\text{enc}}(\mathbf{x})) \approx \mathbf{x}$ for $\mathbf{x} \in \mathcal{X}_{\text{src}} \cup \{\mathbf{x}_{\text{tgt}}\}$, which approximates the distribution-matching aspect of our estimation objectives Equation 2 and Equation 3.

Despite the resemblance on the reconstruction objective, MAE-TTT does not explicitly perform the representation alignment as our objectives – a classifier $f_{\text{cls}}$ is only trained on the labeled source distribution, which takes in $f_{\text{enc}}$'s output $\hat{\mathbf{z}}$ and produces logit values. In addition, our objectives entail maximizing the target likelihood $\hat{p}(\hat{\mathbf{c}}_{\text{tgt}})$ to align $\hat{\mathbf{c}}_{\text{tgt}}$ to the source support $\hat{\mathcal{C}}_{\text{src}}$. As large logit values indicate the sample is close to distribution modes [40, 41] and $f_{\text{enc}}$ is enforced invertible through auto-encoding, we can interpret minimizing the entropy of $f_{\text{cls}}(\hat{\mathbf{z}}_{\text{tgt}})$ as filtering $\hat{\mathbf{z}}_{\text{tgt}}$ to obtain $\hat{\mathbf{c}}_{\text{tgt}}$ and driving it towards the modes of $\hat{p}(\hat{\mathbf{c}})$. Therefore, we implement the entropy minimization loss $\inf - \log \sum_y f_{\text{cls}}(\hat{\mathbf{z}}_{\text{tgt}})_y \log(f_{\text{cls}}(\hat{\mathbf{z}}_{\text{tgt}})_y)$ as a surrogate for maximizing $p(\hat{\mathbf{c}})$. We show that this significantly boosts the performance of MAE-TTT in Section 6.1.

**Regularization.** While our objectives simultaneously involve the source distribution and the target sample, the source distribution may not be accessible during adaptation. Aggressive updates on the target sample may distort the source information stored in the model and ultimately impair the performance. To address this, we propose to impose regularization on the source-pretrained backbone during adaptation to enforce minimal changes and preserve the source information. In Section 6.2, we instantiate this with low-rank updates and sparsity constraints, showcasing the resultant benefits.

## 5   Synthetic Data Experiments

In this section, we conduct synthetic data experiments on classification to directly validate the theoretical results in Section 4. We present additional experiments on regression in Section A4.2.

**Experimental setup.** We generated the synthetic data following the generative process in Equation 1, with $d_{\mathbf{c}} = 4$ and $d_{\mathbf{s}} = 2$. We focus on binary classification and sample class embeddings $\mathbf{c}_1$ and $\mathbf{c}_2$ from $\mathcal{N}(0, \mathbf{I}_c)$ and $\mathcal{N}(2, \mathbf{I}_c)$ respectively. We sample $\mathbf{s}_{\text{src}}$ from a truncated Gaussian centered at the origin and sample $\mathbf{s}_{\text{tgt}}$ at multiple distances from the origin. For the dense-shift case, we concatenate $\mathbf{c}$ and $\mathbf{s}$ and feed them to a well-conditioned 4-layer multi-layer perceptron (MLP) with ReLU activation to obtain $\mathbf{x}$. For the sparse-shift case, we pass $\mathbf{c}$ to a 4-layer MLP to obtain a 4-d vector. We duplicate 2 dimensions of this vector and add $\mathbf{s}$ to it. The final $\mathbf{x}$ is the concatenation of the 4-d vector and the 2-d vector. We sample 10k points for the source distribution and 1 target sample for each run. We perform 50 runs for each configuration and compute the accuracy on the target samples. More details can be found in Appendix A4.

**Results and discussions.** We compared our method with iMSDA [18] and a model trained only on source data. The results in both dense and sparse shift settings are summarized in Table 1. Our method consistently outperforms both baseline methods (nearly random guesses) by a large margin on all sub-settings, validating our theoretical results. The results on iMSDA suggest that directly applying domain-adaptation methods to the extrapolation task may result in negative effects for lack of the target distribution in their training.

## 6   Real-world Data Experiments

We provide real-world experiments to validate our theoretical insights for practical algorithms (Section 4.3) and theoretical results (Section 4.2). More results can be found in Appendix A5. [3]

---

[3]The code is provided here.

Table 2: **Comparison of SOTA TTA Methods on CIFAR10-C, CIFAR100-C, and ImageNet-C.** Average error rates over 15 test corruptions are reported. Baseline results are from Tomar et al. [21]. Values are (means ± standard deviations) over three random seeds. * indicates our reproductions.

| Method | CIFAR10-C | CIFAR100-C | ImageNet-C |
|---|---|---|---|
| Source [21] | 29.1 | 60.4 | 81.8 |
| BN [42] | 15.6 | 43.7 | 67.7 |
| TENT [15] | 14.1 | 39.0 | 57.4 |
| SHOT [43] | 13.9 | 39.2 | 68.7 |
| TTT++ [14] | 15.8 | 44.4 | 59.3 |
| TTAC [44] | 13.4 | 41.7 | 58.7 |
| TeSLA-s [21] | 12.1 | 37.3 | 53.1 |
| **TeSLA-s+SC** | **11.7 ± 0.01 ↓** | **37.0 ± 0.06 ↓** | **50.9 ± 0.15 ↓** |
| TeSLA* [21] | 12.5 ± 0.04 | 38.2 ± 0.03 | 55.0 ± 0.17 |
| **TeSLA+SC** | **12.1 ± 0.11 ↓** | **38.0 ± 0.13 ↓** | **54.5 ± 0.12 ↓** |

## 6.1 Generative Adaptation with Entropy Minimization

As discussed in the first implication in Section 4.3, we incorporate an entropy-minimization loss to MAE-TTT and compare it with the original MAE-TTT.

**Experimental setup.** We conduct experiments on ImageNet-C [45] and ImageNet100-C [46] with 15 different types of corruption. For the baseline, we utilize the publicly available code of MAE-TTT. In our approach, we do not directly integrate the entropy-minimization loss into the MAE-TTT framework. This is because the training process of self-supervised MAE relies on masked images, whereas entropy-minimization requires the classification of the entire image. To address this, we introduce additional training steps with unmasked images and apply the entropy-minimization loss during these steps. Specifically, the training process for each test-time iteration is split into two stages. We first follow the MAE-TTT approach by inputting masked images and training the model using reconstruction loss. In this stage, only the encoder is updated. Then, we input full images (32 in a batch) and optimize the model with the entropy minimization loss following SHOT [43]. In this stage, both the encoder and classifier are optimized. The learning rates for both stages are set the same.

Table 3: **Test accuracy (%) on ImageNet-C.** The baseline results are from Gandelsman et al. [20].

| Acc (%) | brigh | cont | defoc | elast | fog | frost | gauss | glass | impul | jpeg | motn | pixel | shot | snow | zoom | Avg |
|---|---|---|---|---|---|---|---|---|---|---|---|---|---|---|---|---|
| Joint Train | 62.3 | 4.5 | 26.7 | 39.9 | 25.7 | 30.0 | 5.8 | 16.3 | 5.8 | 45.3 | 30.9 | 45.9 | 7.1 | 25.1 | 31.8 | 26.88 |
| Fine-Tune | 67.5 | 7.8 | 33.9 | 32.4 | 36.4 | 38.2 | 22.0 | 15.7 | 23.9 | 51.2 | 37.4 | 51.9 | 23.7 | 37.6 | 37.1 | 34.45 |
| ViT Probe | 68.3 | 6.4 | 24.2 | 31.6 | 38.6 | 38.4 | 17.4 | 18.4 | 18.2 | 51.2 | 32.2 | 49.7 | 18.2 | 35.9 | 32.2 | 32.06 |
| TTT-MAE | 69.1 | 9.8 | **34.4** | 50.7 | 44.7 | 50.7 | 30.5 | 36.9 | 32.4 | 63.0 | **41.9** | 63.0 | 33.0 | 42.8 | **45.9** | 45.92 |
| **Ours** | **73.8** | **14.0** | 33.6 | **69.0** | **47.8** | **64.6** | **38.6** | **42.2** | 36.6 | **68.4** | 32.4 | **67.4** | **41.2** | **51.2** | 35.4 | **47.77** |

**Comparison with baselines.** In Table 3, we compare our method with the baseline MAE-TTT [20] and other baselines therein. We can observe that our algorithm largely boosts the performance of the MAE-TTT baseline over most corruption types. This corroborates our theoretical insights and showcases its practical value.

Table 4: **Understanding entropy-minimization steps on ImageNet100-C**. Values are classification accuracy (mean and standard deviation) over three random seeds.

| | Source | MAE-TTT | Using entropy-minimization | | |
|---|---|---|---|---|---|
| | | | 1 Step | 2 Steps | 3 Steps |
| **Acc** | 50.29 | 59.12 ± 0.35 | 63.99 ± 0.25 | 65.09 ± 0.23 | 65.01 ± 0.39 |

**Understanding entropy-minimization steps.** Table 4 presents the results of entropy-minimization with different training steps. The results indicate that the additional entropy-minimization steps significantly enhance the performance of the MAE-TTT framework, demonstrating the synergy between auto-encoding and entropy-minimization as indicated in our theoretical framework.

## 6.2 Sparsity Regularization

As suggested by the second implication in Section 4.3, we integrate sparsity constraints into the state-of-the-art TTA method, TeSLA/TeSLA-s [21]. Although our theoretical results rely on a generative model, we demonstrate that our implications are also applicable to discriminative models.

**Experimental setup.** We conduct experiments on the CIFAR10-C, CIFAR100-C, and ImageNet-C datasets [45], following the protocols outlined for TeSLA and TeSLA-s [21], with and without training data information. In the pre-train stage, we apply the ResNet50 [47] as the backbone network and follow prior work [14, 44] to pre-train it on the clean CIFAR10, CIFAR100, and ImageNet training sets, with joint contrastive and classification losses. In the test-time adaptation process, we adopt the sequential TTA protocol as outlined in TTAC [44] and TeSLA [21]. This protocol prohibits the change of training objectives throughout the test phase. To encourage sparsity, we add low-rank adaptation (LoRA) modules [48] to the backbone network, which limits the adaptation to low intrinsic dimensions. Beyond LoRA, we further implement a masking layer with corresponding sparsity constraint ($\ell_1$ loss) to filter out redundant changes. More details can be found in Appendix A5.

**Results analysis.** The average error rates under 15 corruption types for all CIFAR10-C, CIFAR100-C, and ImageNet-C datasets are summarized in Table 2. We can observe that sparsity constraints consistently improve performance over the current SOTA method, TeSLA/TeSLA-s, across all three datasets. The lightweight nature of the sparsity constraint and its consistent performance enhancements make it a valuable addition. This demonstrates the potential of sparsity constraints as a versatile, plug-and-play module for enhancing existing TTA methods.

## 6.3 Shift Scope and Severity

To investigate the trade-off between the shift scope (dense vs. sparse) and severity, we simulate different levels of corruption severity and corrupted region sizes and evaluate a classical TTA method TENT [15] on these configurations. Following [45], we inject impulse noise to the CIFAR10 dataset, with noise levels ranging from 1 to 10 to simulate various severity levels. To control the shift's scope, we crop regions of various sizes and introduce corruption only to this region. Figure 3 displays classification error curves under various shift severity levels and region sizes. We can observe that classification errors rise with increasing noise levels and region sizes. Notably, for large block sizes (dense shifts), the performance dramatically declines and even collapses as the severity level rises, whereas the performance remains almost constant over all severity levels in the sparse shift regime, verifying the theoretical conditions for Theorem 4.2 and Theorem 4.4.

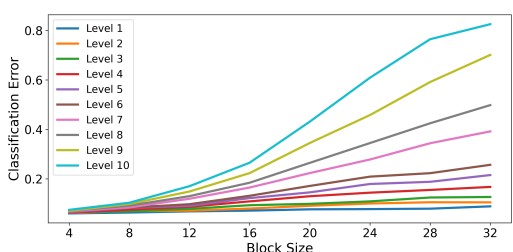

Figure 3: **TTA classification errors under different levels of shift severity levels and scopes.**

## 7 Conclusion and Limitations

In this work, we characterize extrapolation with a latent-variable model that encodes a minimal change principle. Within this framework, we establish clear conditions under which extrapolation becomes not only feasible but also guaranteed, even for complex nonlinear models in deep learning. Our conditions reveal the intricate interplay among the generating function's smoothness, the out-of-support degree, and the influence of the shift. These theoretical results provide valuable implications for the design of practical test time adaptation methods, which we validate empirically.

**Limitations**: On the theory aspect, the Jacobian norm utilized in Theorem 4.2 only considers the global smoothness of the generating function and thus may be too stringent if the function is much more well-behaved/smooth over the extrapolation region of concern. Therefore, one may consider a refined local condition to relax this condition. On the empirical side, our theoretical framework entails learning an explicit representation space. Existing methods without such a structure may still benefit from our framework but to a lesser extent. Also, our framework involves several loss terms including reconstruction, classification, and the likelihood of the target invariant variable. A careful re-weighting of these terms may be needed during training.

**Acknowledgments.** We thank the anonymous reviewers for their valuable insights and recommendations, which have greatly improved our work. The work of L. Kong is supported in part by NSF DMS-2134080 through an award to Y. Chi. This material is based upon work supported by NSF Award No. 2229881, AI Institute for Societal Decision Making (AI-SDM), the National Institutes of Health (NIH) under Contract R01HL159805, and grants from Salesforce, Apple Inc., Quris AI, and Florin Court Capital. P. Stojanov was supported in part by the National Cancer Institute (NCI) grant number: K99CA277583-01, and funding from the Eric and Wendy Schmidt Center at the Broad Institute of MIT and Harvard. This research has been graciously funded by the National Science Foundation (NSF) CNS2414087, NSF BCS2040381, NSF IIS2123952, NSF IIS1955532, NSF IIS2123952; NSF IIS2311990; the National Institutes of Health (NIH) R01GM140467; the National Geospatial Intelligence Agency (NGA) HM04762010002; the Semiconductor Research Corporation (SRC) AIHW award 2024AH3210; the National Institute of General Medical Sciences (NIGMS) R01GM140467; and the Defense Advanced Research Projects Agency (DARPA) ECOLE HR00112390063. Any opinions, findings, and conclusions or recommendations expressed in this publication are those of the author(s) and do not necessarily reflect the views of the National Science Foundation, the National Institutes of Health, the National Geospatial Intelligence Agency, the Semiconductor Research Corporation, the National Institute of General Medical Sciences, and the Defense Advanced Research Projects Agency.

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

*Appendix for*

**"Towards Understanding Extrapolation: a Causal Lens "**

Table of Contents

## A1 Related Work

In this section, we discuss some related topics including extrapolation, latent-variable identification, and test-time adaptation.

**Extrapolation.** Out-of-distribution generalization has attracted significant attention in recent years. Unlike our work, the bulk of the work is devoted to generalizing to target distributions on the same support as the source distribution [22, 23, 8]. Recent work [24–27] investigates extrapolation in the form of compositional generalization by resorting to structured generating functions (e.g., additive, slot-wise). Another line of work [28–30] studies extrapolation in regression problems and does not consider the latent representation. Saengkyongam et al. [31] leverage a latent variable model and assumes a linear relation between the intervention variable and the latent variable to handle extrapolation. In this work, we formulate extrapolation as a latent variable identification problem. Unlike the semi-parametric conditions in prior work, our conditions do not constrain the form of the generating function and are more compatible with deep learning models and tasks. We demonstrate that our conditions naturally lead to implications benefiting practical deep-learning algorithms.

**Latent-variable identification for transfer learning.** Identifying latent variables in a causal model has become one canonical paradigm to formalize and understand representation learning in the deep learning regime. Typically, one would assume some latent variables $\mathbf{z}$ generate the observed data $\mathbf{x}$ (e.g., images, text) through a generating function. However, the nonlinearity of deep learning models requires the generating function to be nonlinear, which has posed major technical difficulty in recovering the original latent variable [32]. To overcome this setback, a line of work [33–36] assumes the availability of an auxiliary label $\mathbf{u}$ for each sample $\mathbf{x}$ and under different $\mathbf{u}$ values, each component $z_i$ of $\mathbf{z}$ experiences sufficiently large shift in its distribution. This condition leads to component-wise identification of $\mathbf{z}$, i.e., each estimate $\hat{z}_i$ is equivalent to $z_{\pi(i)}$ up to an invertible mapping for a permutation function $\pi : [d_z] \rightarrow [d_z]$. Since this framework assumes all latent components' distributions vary over distributions indexed by $\mathbf{u}$, it doesn't assume the existence of some shared, invariant information across distributions, which is often the case for transfer learning tasks. To address this issue, recent work [18, 19] introduce a partition of $\mathbf{z}$ into an invariable variable $\mathbf{c}$ and an changing variable $\mathbf{s}$ (i.e., $\mathbf{z} := [\mathbf{c}, \mathbf{s}]$) such that $\mathbf{c}$'s distribution remains constant over distributions. They show both $\mathbf{c}$ and $\mathbf{s}$ can be identified and one can directly utilize the invariant variable $\mathbf{c}$ for domain adaptation. However, their techniques crucially rely on the variability of the changing variable $\mathbf{s}$, mandating the availability of multiple sufficiently disparate distributions (including the target) and their overlapping supports. These constraints make them unsuitable for the extrapolation problem. In comparison, our theoretical results give identification of the invariant variable $\mathbf{c}$ (the on-support variable in the extrapolation context) with only one source distribution $p_{\mathrm{src}}(\mathbf{x})$ and as few as one out-off support target sample $\mathbf{x}_{\mathrm{tgt}}$ through mild assumptions on the generating function, which directly tackles the extrapolation problem.

**Test-time adaptation.** Test-time Adaptation (TTA) aims at adapting models trained on a source domain to align with the target domain during testing [49–55]. It is broadly classified based on whether the training objective is modified. Test-time Training (TTT) methods [13, 14, 44, 56, 57], including TTT [13] and TTT++ [14], proficiently adjust models to target domains by implementing similar self-supervised learning strategies on both training and testing data. In contrast, Sequential Test-Time Adaptation [15, 54, 55, 58–63] (sTTA) garners significant interest due to its practicality, notably its one-pass sequential inference and no training objective access. Research in sTTA primarily concentrates on two facets: the selection of model parameters for adaptation and the refinement of pseudo-labeling techniques for enhanced efficiency. For instance, TENT [15] fine-tunes the Batch Normalization (BN) layers by minimizing entropy, SHOT [16] adjusts the backbone network while maintaining a static classifier, and T3A [64] updates the classifier prototype. Moreover, a burgeoning line of research [65, 21, 44, 50, 15, 16] focuses on deriving more robust self-training signals through improved pseudo labeling strategies. For example, TTAC [44] employs clustering techniques to extract more accurate pseudo labels. Despite the prominent recent development, these algorithms tend to be brittle and sensitive to hyper-parameter tuning [66] and limited in theoretical understanding [17]. Our work offers formalization and understanding to fill in this gap. We show that insights inferred from our theory can indeed benefit existing TTA algorithms, which hopefully will serve as the first step to bridge the theory and practice for TTA algorithms.

## A2 Proof for Theorem 4.2

**Assumption 4.1** (Identification Conditions under Global Shifts)**.**

  i *[Smoothness & Invertibility]: The generating function g in Equation 1 is a smooth invertible function with a smooth inverse everywhere.*

  ii *[Compactness]: The source data space $\mathcal{X}_{\mathrm{src}} \subset \mathbb{R}^{d_x}$ is closed and bounded.*

  iii *[Discreteness]: The invariant variable $\mathbf{c}$ takes on values from a finite set: $\mathcal{C} = \{\mathbf{c}_k\}_{k \in [K]}$.*

  iv *[Continuity]: The probability density function $p(\mathbf{s}|\mathbf{c})$ is continuous over $\mathbf{s} \in \mathcal{S}_{\mathrm{src}}$, for all $\mathbf{c} \in \mathcal{C}$.*

  v *[Out-of-support Distance]: The target sample's out-support components $\mathbf{s}_{\mathrm{tgt}}$'s distance from the source support $\mathcal{S}_{src}$ is constrained: $\inf_{\mathbf{s} \in \mathcal{S}_{src}} \|\mathbf{s}_{\mathrm{tgt}} - \mathbf{s}\| \leq \frac{\min_{\mathbf{c} \in \mathcal{C} \setminus \{\mathbf{c}_{\mathrm{tgt}}\}} D(\mathbf{c}_{\mathrm{tgt}}, \mathbf{c})}{2 J_u}.$*

We first present Lemma A1 from Kong et al. [38] which establishes the discrete information on the source support and serves as the starting point in the proof of Theorem 4.2.

**Lemma A1** (Source discrete subspace identification [38])**.** *Assuming a generating process in Equation 1, we estimate the distribution with model $(\hat{g}, \hat{p}(\hat{\mathbf{c}}), \hat{p}(\hat{\mathbf{s}}))$. Under Assumption 4.1 i,ii,iii,iv, it follows that the estimated variable $\hat{\mathbf{c}}$ takes on values from $\{\hat{\mathbf{c}}_k\}_{k=1}^{K}$ where each value corresponds uniquely to one value of the true variable $\mathbf{c}$, i.e., $\mathbf{c} = \mathbf{c}_k \iff \hat{\mathbf{c}} = \hat{\mathbf{c}}_k$.*

**Theorem 4.2** (Extrapolation under Dense Shifts)**.** *Assuming a generating process in Equation 1, we estimate the distribution with model $(\hat{g}, \hat{p}(\hat{\mathbf{c}}), \hat{p}(\hat{\mathbf{s}}))$ with the objective:*

$$\sup \hat{p}(\hat{\mathbf{c}}_{\mathrm{tgt}}), \quad \textit{Subject to: } \hat{p}(\mathbf{x}) = p(\mathbf{x}), \forall \mathbf{x} \in \mathcal{X}_{\mathrm{src}}; \quad \hat{\mathbf{s}}_{\mathrm{tgt}} \in \arg\inf_{\hat{\mathbf{s}}} D(\hat{\mathbf{s}}, \hat{\mathcal{S}}_{\mathrm{src}}). \quad (2)$$

*Under Assumption 4.1, the estimated model can attain the identifiability in Definition 3.1.*

*Proof for Theorem 4.2.* Lemma A1 shows that the discrete invariant variable $\mathbf{c}$ is identifiable on the source distribution.

In the following, we show that the target's invariant variable $\mathbf{c}_{\mathrm{tgt}}$ is identifiable if $\mathbf{s}_{\mathrm{tgt}}$ does not drift too far away from the source support $\mathcal{S}_{\mathrm{src}}$. Suppose that $\mathbf{x}_{\mathrm{tgt}}$ resides on both manifolds $g(\mathbf{c}_k, \cdot)$ and $g'(\mathbf{c}_{k'}, \cdot)$ where $k \neq k'$. The generating function $g' \in \mathcal{G}$ belongs to the generating function class and behaves exactly the same as $g$ on the source support, i.e., $g' = g$ over $\mathcal{C} \times \mathcal{S}_{\mathrm{src}}$. We define the minimal distance $D(\mathbf{c}_k, \mathbf{c}_{k'})$ between the two manifolds on support boundaries, i.e., $D(\mathbf{c}_k, \mathbf{c}_{k'}) := \min_{\mathbf{s}_1, \mathbf{s}_2 \in \mathrm{Bd}(\mathcal{S}_{\mathrm{src}})} \|g(\mathbf{c}_k, \mathbf{s}_1) - g(\mathbf{c}_{k'}, \mathbf{s}_2)\| > 0$. Since $\mathbf{x}_{\mathrm{tgt}}$ lives on both manifolds $g(\mathbf{c}_k, \cdot)$ and $g'(\mathbf{c}_{k'}, \cdot)$, we can express it as $\mathbf{x}_{\mathrm{tgt}} = g(\mathbf{c}_k, \mathbf{s}_{\mathrm{tgt}}) = g'(\mathbf{c}_{k'}, \mathbf{s}'_{\mathrm{tgt}})$. We define $\mathbf{s}_{\mathrm{src}} \in \arg\min_{\mathbf{s} \in \mathcal{S}_{\mathrm{src}}} \|\mathbf{s} - \mathbf{s}_{\mathrm{tgt}}\|$ and $\mathbf{s}'_{\mathrm{src}} \in \arg\min_{\mathbf{s} \in \mathcal{S}_{\mathrm{src}}} \|\mathbf{s} - \mathbf{s}'_{\mathrm{tgt}}\|$ as two closest points on the source support to $\mathbf{s}_{\mathrm{tgt}}$ and $\mathbf{s}'_{\mathrm{tgt}}$ respectively. It follows that

$$\mathbf{x}_{\mathrm{tgt}} - g(\mathbf{c}_k, \mathbf{s}_{\mathrm{src}}) = \left( \int_0^1 \mathbf{J}_{g(\mathbf{c}_k, \cdot)}(\mathbf{s}_{\mathrm{src}} + t \cdot \mathbf{h}) dt \right) \mathbf{h};$$

$$\mathbf{x}_{\mathrm{tgt}} - g'(\mathbf{c}_{k'}, \mathbf{s}'_{\mathrm{src}}) = \mathbf{x}_{\mathrm{tgt}} - g(\mathbf{c}_{k'}, \mathbf{s}'_{\mathrm{src}}) = \left( \int_0^1 \mathbf{J}_{g(\mathbf{c}_{k'}, \cdot)}(\mathbf{s}'_{\mathrm{src}} + t \cdot \mathbf{h}') dt \right) \mathbf{h}', \quad (4)$$

where $\mathbf{h} := \mathbf{s}_{\mathrm{tgt}} - \mathbf{s}_{\mathrm{src}}$ and $\mathbf{h}' := \mathbf{s}'_{\mathrm{tgt}} - \mathbf{s}'_{\mathrm{src}}$.

It follows from Equation 4

$$g(\mathbf{c}_{k'}, \mathbf{s}'_{\text{src}}) - g(\mathbf{c}_k, \mathbf{s}_{\text{src}}) = \left( \int_0^1 \mathbf{J}_{g(\mathbf{c}_k, \cdot)}(\mathbf{s}_{\text{src}} + t \cdot \mathbf{h}) dt \right) \mathbf{h} - \left( \int_0^1 \mathbf{J}_{g(\mathbf{c}_{k'}, \cdot)}(\mathbf{s}'_{\text{src}} + t \cdot \mathbf{h}') dt \right) \mathbf{h}';$$

$$\Longrightarrow$$

$$\left\| \left( \int_0^1 \mathbf{J}_{g(\mathbf{c}_k, \cdot)}(\mathbf{s}_{\text{src}} + t \cdot \mathbf{h}) dt \right) \mathbf{h} - \left( \int_0^1 \mathbf{J}_{g(\mathbf{c}_{k'}, \cdot)}(\mathbf{s}'_{\text{src}} + t \cdot \mathbf{h}') dt \right) \mathbf{h}' \right\| \geq D(\mathbf{c}_k, \mathbf{c}_{k'});$$

$$\Longrightarrow$$

$$\left\| \left( \int_0^1 \mathbf{J}_{g(\mathbf{c}_k, \cdot)}(\mathbf{s}_{\text{src}} + t \cdot \mathbf{h}) dt \right) \mathbf{h} \right\| + \left\| \left( \int_0^1 \mathbf{J}_{g(\mathbf{c}_{k'}, \cdot)}(\mathbf{s}'_{\text{src}} + t \cdot \mathbf{h}') dt \right) \mathbf{h}' \right\| \geq D(\mathbf{c}_k, \mathbf{c}_{k'});$$

$$\Longrightarrow$$

$$J_{\text{u}}(\|\mathbf{h}\| + \|\mathbf{h}'\|) \geq D(\mathbf{c}_k, \mathbf{c}_{k'});$$

$$\Longrightarrow$$

$$\max\{\|\mathbf{h}\|, \|\mathbf{h}'\|\} \geq \frac{D(\mathbf{c}_k, \mathbf{c}_{k'})}{2J_{\text{u}}}.$$

(5)

Assumption 4.1- v states that $\|\mathbf{h}\| < \frac{D(\mathbf{c}_k, \mathbf{c}_{k'})}{2J_{\text{u}}}$ for the true generating function $g$. Therefore, $\mathbf{x}_{\text{tgt}}$ can only be explained by one manifold, which we denote as $g(\mathbf{c}_{\text{tgt}}, \cdot)$.

Finally, we show that the objective Equation 2 guarantees that the solution $\hat{\mathbf{c}}_{\text{tgt}}$ corresponds to the true $\mathbf{c}_{\text{tgt}}$. We suppose that $\mathbf{c}_{\text{tgt}} = \mathbf{c}_k$ which corresponds to $\hat{\mathbf{c}}_k$ for a specific $k \in [K]$. First, we note that $\hat{\mathbf{c}}_{\text{tgt}}$ could only take values from $\{\hat{\mathbf{c}}_k\}_{k \in [K]}$ due to the constraint $\sup \hat{p}(\hat{\mathbf{c}}_{\text{d}})$. Also, the correct solution $\hat{\mathbf{c}}_k$ is always a feasible solution to the objective Equation 2, since $\hat{g}$ can take on the true generating function $g$. Thus, for another plausible solution $\hat{\mathbf{c}}_{k'} \neq \hat{\mathbf{c}}_k$, we would have

$$\max\{\|\hat{\mathbf{h}}\|, \|\hat{\mathbf{h}}'\|\} \geq \frac{D(\hat{\mathbf{c}}_k, \hat{\mathbf{c}}_{k'})}{2\hat{J}_{\text{u}}},$$

(6)

where the definitions are analogous to those in Equation 5 and decorated with $\hat{\cdot}$ to indicate the difference. Due to the distance-minimization term $\min_{\hat{g}, \hat{\mathbf{s}}_{\text{src}} \in \hat{\mathcal{S}}_{\text{src}}} \|\hat{\mathbf{s}}_{\text{tgt}} - \hat{\mathbf{s}}_{\text{src}}\|$, the distance for the correct solution $\hat{\mathbf{c}}_k$ is upper-bounded by $\|\hat{\mathbf{h}}\| < \frac{D(\hat{\mathbf{c}}_k, \hat{\mathbf{c}}_{k'})}{2\hat{J}_{\text{u}}}$, since this is attainable when the estimated generating function is the true function, i.e., $g = \hat{g}$. Equation 6 implies that the alternative solution $\hat{\mathbf{c}}_{k'}$ would always yield $\|\hat{\mathbf{h}}'\| \geq \frac{D(\hat{\mathbf{c}}_k, \hat{\mathbf{c}}_{k'})}{2\hat{J}_{\text{u}}} > \|\hat{\mathbf{h}}\|$, which the distance-minimizing regularization would exclude. Therefore, we have shown that the estimated $\hat{\mathbf{c}}_{\text{tgt}}$ corresponds to the correct $\mathbf{c}_k$. $\square$

## A3  Proof for Theorem 4.4

**Assumption 4.3** (Identification Conditions under Local Shifts)**.**

   i *[Smoothness & Invertibility]: The generating function $g$ in Equation 1 is invertible and differentiable, and its inverse is also differentiable.*

   ii *[Invariant Variable Informativeness]: The dimensions under $\mathbf{c}$'s exclusive influence is uniquely determined: for a fixed $\mathbf{c} \in \mathcal{C}$, $[\mathbf{x}]_{\mathcal{I}_{\mathbf{c} \setminus \mathbf{s}}(\mathbf{c}, \mathbf{s}_1)} \neq [\mathbf{x}]_{\mathcal{I}_{\mathbf{c} \setminus \mathbf{s}}(\mathbf{c}^*, \mathbf{s}_2)}$ for any $\mathbf{c}^* \neq \mathbf{c}$, $\mathbf{s}_1 \in \mathcal{S}$, and $\mathbf{s}_2 \in \mathcal{S}$.*

   iii *[Sparse Influence]: At any $\mathbf{z} \in \mathcal{Z}$, the changing variable $\mathbf{s}$ influences at most $d_{\mathbf{s}}$ dimensions of $\mathbf{x}$, i.e., $|\mathcal{I}_{\mathbf{s}}(\mathbf{z})| \leq d_{\mathbf{s}}$. Alternatively, the two variables $\mathbf{c}$ and $\mathbf{s}$ do not intersect on their influenced dimensions $\mathcal{I}_{\mathbf{c}}(\mathbf{z}) \cap \mathcal{I}_{\mathbf{s}}(\mathbf{z}) = \emptyset$.*

   iv *[Mechanistic Dependence]: For all $\mathbf{z}$, any nontrivial partition $\mathcal{P}_1, \mathcal{P}_2$ of the dimensions $\mathcal{I}_{\mathbf{c} \setminus \mathbf{s}}(\mathbf{z})$ yields dependence between the sub-matrices of the Jacobian $\mathbf{J}_g(\mathbf{z})$: $rank([\mathbf{J}_g(\mathbf{z})]_{\mathcal{I}_{\mathbf{c} \setminus \mathbf{s}}}(\mathbf{z})) < rank([\mathbf{J}_g(\mathbf{z})]_{\mathcal{P}_1}(\mathbf{z})) + rank([\mathbf{J}_g(\mathbf{z})]_{\mathcal{P}_2}(\mathbf{z}))$.*

**Theorem 4.4** (Extrapolation under Sparse Shifts)**.** *Assuming a generating process in Equation 1, we estimate the distribution with model $(\hat{g}, \hat{p}(\hat{\mathbf{c}}), \hat{p}(\hat{\mathbf{s}}))$ with the objective:*

$$\sup \hat{p}(\hat{\mathbf{c}}_{\text{tgt}}), \quad \textit{Subject to:} \quad \hat{p}(\mathbf{x}) = p(\mathbf{x}), \forall \mathbf{x} \in \mathcal{X}_{\text{src}}.$$

(3)

*Under Assumption 4.3, the estimated model can attain the identifiability in Definition 3.1.*

**Lemma A1** (Brady et al. [39])**.** *Let $g, \hat{g} : \mathbb{R}^{d_{\mathbf{z}}} \to \mathbb{R}^{d_{\mathbf{x}}}$ be smooth and invertible. Then, for any $\mathbf{z} \in \mathbb{R}^{\mathbf{z}}$, $\mathcal{S} \subset \mathbb{R}^{d_{\mathbf{s}}}$, $rank([\mathbf{J}_g(\mathbf{z})]_{\mathcal{S}}) = rank([\mathbf{J}_{\hat{g}}(\hat{\mathbf{z}})]_{\mathcal{S}})$, where $\hat{\mathbf{z}} := \hat{g}^{-1} \circ g(\mathbf{z})$.*

*Proof.* The proof consists of two steps. In the first step, we show the identification of the index set $\mathcal{I}_{\mathbf{c} \backslash \mathbf{s}}(\mathbf{z}) \subset [d_{\mathbf{x}}]$ over which $\mathbf{x}$ receives only $\mathbf{c}$' influence. That is, $\hat{g}$ maps the estimated invariant variable $\hat{\mathbf{c}}$ to $\mathbf{x}$ dimensions generated by the true invariant variable $\mathbf{c}$ alone. In the second step, we show that the objective $\sup \hat{p}(\hat{\mathbf{c}}_{\text{tgt}})$ assigns to the estimated invariant variable for the target sample $\hat{\mathbf{c}}_{\text{tgt}}$ the source-distribution estimated value $\hat{\mathbf{c}}_{\text{src}}$ that is also generated by $\mathbf{c}_{\text{tgt}}$. Recall the notation $\mathbf{z} := [\mathbf{c}, \mathbf{s}]$. We denote the latent source support as $\mathcal{Z}_{\text{src}}$ and the set augmented with the target sample as $\mathcal{Z} := \mathcal{Z}_{\text{src}} \cup \{\mathbf{z}_{\text{tgt}}\}$

**Step 1.** We first show that $\hat{g}$ cannot map the estimated invariant variable $\hat{\mathbf{c}}$ to $\mathbf{x}$ dimensions generated by both the invariant variable $\mathbf{c}$ and the changing variable $\mathbf{s}$.

We show this by contradiction. Suppose that $\exists \mathbf{z}^* \in \mathcal{Z}$, such that

$$\hat{\mathcal{I}}_{\mathbf{c} \backslash \mathbf{s}}(\hat{\mathbf{z}}^*) \cap \mathcal{I}_{\mathbf{c} \backslash \mathbf{s}}(\mathbf{z}^*) \neq \emptyset \text{ and } \hat{\mathcal{I}}_{\mathbf{c} \backslash \mathbf{s}}(\hat{\mathbf{z}}^*) \cap \mathcal{I}_{\mathbf{s}}(\mathbf{z}^*) \neq \emptyset \tag{7}$$

We partition $\mathcal{I}_{\mathbf{c} \backslash \mathbf{s}}(\mathbf{z}^*)$ into two disjoint sets $\mathcal{I}_{\mathbf{c} \backslash \mathbf{s}, 1} := \{i \in \mathcal{I}_{\mathbf{c} \backslash \mathbf{s}}(\mathbf{z}^*) | i \in \hat{\mathcal{I}}_{\mathbf{c} \backslash \mathbf{s}}(\hat{\mathbf{z}}^*)\}$ and $\mathcal{I}_{\mathbf{c} \backslash \mathbf{s}, 2} := \{i \in \mathcal{I}_{\mathbf{c} \backslash \mathbf{s}}(\mathbf{z}^*) | i \notin \hat{\mathcal{I}}_{\mathbf{c} \backslash \mathbf{s}}(\hat{\mathbf{z}}^*)\}$ base on the overlap with $\hat{\mathcal{I}}_{\mathbf{c} \backslash \mathbf{s}}(\hat{\mathbf{z}}^*)$. By the supposed condition Equation 7, it follows that $\mathcal{I}_{\mathbf{c} \backslash \mathbf{s}, 1} \neq \emptyset$ and $\mathcal{I}_{\mathbf{s}, 1} \neq \emptyset$.

We now show that $\mathcal{I}_{\mathbf{c} \backslash \mathbf{s}, 2}$ is nonempty by contradiction. Suppose that $\mathcal{I}_{\mathbf{c} \backslash \mathbf{s}, 2}$ is empty. It follows that $\mathcal{I}_{\mathbf{c} \backslash \mathbf{s}} = \mathcal{I}_{\mathbf{c} \backslash \mathbf{s}, 2} \subset \hat{\mathcal{I}}_{\mathbf{c} \backslash \mathbf{s}}$. By definition, we know that $\mathcal{I}_{\mathbf{s}, 1} \subset \hat{\mathcal{I}}_{\mathbf{c} \backslash \mathbf{s}}$. Thus, $A := \mathcal{I}_{\mathbf{c} \backslash \mathbf{s}} \cup \mathcal{I}_{\mathbf{s}, 1} \subset \hat{\mathcal{I}}_{\mathbf{c} \backslash \mathbf{s}}$. This inclusion implies that

$$\text{rank}([\mathbf{J}_g(\mathbf{z}^*)]_{A,:}) = \text{rank}([\mathbf{J}_{\hat{g}}(\hat{\mathbf{z}}^*)]_{A,:}) \leq d_{\mathbf{c}}. \tag{8}$$

Since $\mathcal{I}_{\mathbf{c} \backslash \mathbf{s}} \cap \mathcal{I}_{\mathbf{s}, 1} = \emptyset$, $[\mathbf{J}_g(\mathbf{z}^*)]_{\mathcal{I}_{\mathbf{c} \backslash \mathbf{s}}, d_{\mathbf{c}}+1:d_{\mathbf{z}}} = 0$, and each row of $[\mathbf{J}_g(\mathbf{z}^*)]_{\mathcal{I}_{\mathbf{s}}, d_{\mathbf{c}}+1:d_{\mathbf{z}}}$ is nonzero by definition, we have that

$$\text{rank}([\mathbf{J}_g(\mathbf{z}^*)]_{A,:}) > \text{rank}([\mathbf{J}_g(\mathbf{z}^*)]_{\mathcal{I}_{\mathbf{c} \backslash \mathbf{s}},:}) = d_{\mathbf{c}}, \tag{9}$$

which contradicts Equation 8. Therefore, $\mathcal{I}_{\mathbf{c} \backslash \mathbf{s}, 2}$ is nonempty. Since we have rank$([\mathbf{J}_g(\mathbf{z}^*)]_{\mathcal{I}_{\mathbf{c} \backslash \mathbf{s}}}) = d_{\mathbf{c}}$ and $(\mathcal{I}_{\mathbf{c} \backslash \mathbf{s}, 1}, \mathcal{I}_{\mathbf{c} \backslash \mathbf{s}, 2})$ forms a pair of nonempty partition, Assumption 4.3-iv implies that

$$\text{rank}([\mathbf{J}_g(\mathbf{z}^*)]_{\mathcal{I}_{\mathbf{c} \backslash \mathbf{s}, 1},:}) + \text{rank}([\mathbf{J}_g(\mathbf{z}^*)]_{\mathcal{I}_{\mathbf{c} \backslash \mathbf{s}, 2},:}) > \text{rank}([\mathbf{J}_g(\mathbf{z}^*)]_{\mathcal{I}_{\mathbf{c} \backslash \mathbf{s}},:}) = d_{\mathbf{c}}. \tag{10}$$

As the $g$'s and $\hat{g}$'s Jacobian matrix ranks are related (Lemma A1), Equation 10 implies that

$$\text{rank}([\mathbf{J}_{\hat{g}}(\hat{\mathbf{z}}^*)]_{\mathcal{I}_{\mathbf{c} \backslash \mathbf{s}, 1},:}) + \text{rank}([\mathbf{J}_{\hat{g}}(\hat{\mathbf{z}}^*)]_{\mathcal{I}_{\mathbf{c} \backslash \mathbf{s}, 2},:}) > d_{\mathbf{c}}. \tag{11}$$

Since $\mathcal{I}_{\mathbf{c} \backslash \mathbf{s}, 1} \subset \hat{\mathcal{I}}_{\mathbf{c} \backslash \mathbf{s}, 1}$ and $\mathcal{I}_{\mathbf{c} \backslash \mathbf{s}, 2} \cap \hat{\mathcal{I}}_{\mathbf{c} \backslash \mathbf{s}, 1} = \emptyset$ by definition and $[\mathbf{J}_{\hat{g}}(\hat{\mathbf{z}}^*)]_{\mathcal{I}_{\mathbf{c} \backslash \mathbf{s}, 2}, d_{\mathbf{c}}+1:d_{\mathbf{z}}}$ has a full-row rank (Assumption 4.3-iii), it follows that

$$\text{rank}([\mathbf{J}_{\hat{g}}(\hat{\mathbf{z}}^*)]_{\mathcal{I}_{\mathbf{c} \backslash \mathbf{s}},:}) = \text{rank}([\mathbf{J}_{\hat{g}}(\hat{\mathbf{z}}^*)]_{\mathcal{I}_{\mathbf{c} \backslash \mathbf{s}, 1},:}) + \text{rank}([\mathbf{J}_{\hat{g}}(\hat{\mathbf{z}}^*)]_{\mathcal{I}_{\mathbf{c} \backslash \mathbf{s}, 2},:}) > d_{\mathbf{c}}. \tag{12}$$

However, Lemma A1 implies that

$$\text{rank}([\mathbf{J}_{\hat{g}}(\hat{\mathbf{z}}^*)]_{\mathcal{I}_{\mathbf{c} \backslash \mathbf{s}},:}) = \text{rank}([\mathbf{J}_g(\mathbf{z}^*)]_{\mathcal{I}_{\mathbf{c} \backslash \mathbf{s}},:}) = d_{\mathbf{c}}. \tag{13}$$

Thus, we have arrived at a contradiction. We have shown that $\hat{g}$ maps the estimated invariant variable $\hat{\mathbf{c}}$ to $\mathbf{x}$ dimensions generated by the true invariant variable $\mathbf{c}$ alone, i.e., $\hat{\mathcal{I}}_{\mathbf{c} \backslash \mathbf{s}}(\hat{\mathbf{z}}) \subset \mathcal{I}_{\mathbf{c} \backslash \mathbf{s}}(\mathbf{z})$. Moreover, if an index $i'$ from the $\hat{\mathbf{s}}$'s region $\hat{\mathcal{I}}_{\mathbf{s}}(\hat{\mathbf{z}})$ also belonged to $\mathcal{I}_{\mathbf{c} \backslash \mathbf{s}}(\mathbf{z})$, i.e., $i' \in \hat{\mathcal{I}}_{\mathbf{s}}(\hat{\mathbf{z}}) \cap \mathcal{I}_{\mathbf{c} \backslash \mathbf{s}}(\mathbf{z})$, then we would have rank$([\mathbf{J}_g(\mathbf{z})]_{\mathcal{I}_{\mathbf{c} \backslash \mathbf{s}}(\mathbf{z}),:}) = \text{rank}([\mathbf{J}_{\hat{g}}(\hat{\mathbf{z}})]_{\mathcal{I}_{\mathbf{c} \backslash \mathbf{s}}(\mathbf{z}),:}) > \text{rank}([\mathbf{J}_{\hat{g}}(\hat{\mathbf{z}})]_{\hat{\mathcal{I}}_{\mathbf{c} \backslash \mathbf{s}}(\hat{\mathbf{z}}),:}) = d_{\mathbf{c}}$. Thus, it follows that we can identify the index set exclusively influenced by $\mathbf{c}$ over $\mathbf{z} \in \mathcal{Z}$:

$$\mathcal{I}_{\mathbf{c} \backslash \mathbf{s}}(\mathbf{z}) = \hat{\mathcal{I}}_{\mathbf{c} \backslash \mathbf{s}}(\hat{\mathbf{z}}). \tag{14}$$

**Step 2.** By definition, each value of $\hat{\mathbf{c}}$ determines the region $[\mathbf{x}]_{\hat{\mathcal{I}}_{\mathbf{c} \backslash \mathbf{s}}(\hat{\mathbf{c}}, \hat{\mathbf{s}})}$. Due to Equation 14, we have $[\mathbf{x}]_{\hat{\mathcal{I}}_{\mathbf{c} \backslash \mathbf{s}}(\hat{\mathbf{c}}, \hat{\mathbf{s}})} = [\mathbf{x}]_{\mathcal{I}_{\mathbf{c} \backslash \mathbf{s}}(\mathbf{c}, \mathbf{s})}$, where $\mathbf{c}$ can only take on a unique value according to Assumption 4.3-ii. Therefore, there exists an one-to-one mapping $h_{\hat{\mathbf{c}}}$ from $\hat{\mathbf{c}}$ to $\mathbf{c}$ over $\mathcal{Z}$. Further, since the maximum likelihood estimation is performed over the entire source distribution $p(\mathbf{x})$ (Equation 3), the image of $h_{\hat{\mathbf{c}}}$ equal to the entire $\mathcal{C}$, thus is onto. Thus, we have shown that there exists an invertible mapping $h_{\hat{\mathbf{c}}} : \hat{\mathbf{c}} \mapsto \mathbf{c}$ valid over $\mathcal{Z}$ (including the target sample), resulting from our objective (Equation 3).

$\square$

Table A1: **Synthetic data results on regression** (MSE) under both dense and sparse shifts across various distances.

| Shifts | Dense | | | Sparse | | |
|---|---|---|---|---|---|---|
| Distance | 18 | 24 | 30 | 18 | 24 | 30 |
| Only Source | 11.64 | 2.44 | 3.26 | 1.84 | 3.32 | 5.84 |
| **Ours** | **1.40** | **1.60** | **1.68** | **1.15** | **1.48** | **1.60** |

## A4 Synthetic Data Experiments

### A4.1 Implementation Details

We employ a variational auto-encoder [67] whose encoder and decoder are both 4-layer MLP with 32 dimensions and leaky ReLu ($\alpha = 0.2$). Following Equation 2 and Equation 3, we implement reconstruction loss, KL loss on the source distribution, the likelihood loss on the target sample, and a classification loss on the source data. For the dense case, we implement an additional distance loss to minimize the $\ell_2$ distance of $\hat{\mathbf{s}}_{\text{tgt}}$ to the center of the source support (which is the origin in our case). The source-only baseline is trained only with classification loss. The iMSDA implementation is adopted directly from the source code of Kong et al. [18]. We train all methods with Adam [68] and learning rate $2e - 3$ for 25 epochs. We fix the loss weights $\lambda_{\text{cls}} = 1$, $\lambda_{\text{recons}} = 0.1$, $\lambda_{\text{tgt\_likelihood}} = 0.1$, and $\lambda_{\text{s\_distance}} = 0.01$ (for dense shifts) overall distance configurations. We only tune $\lambda_{\text{KL}}$ from the interval $\{1e - 1, 1e - 2, 1e - 3\}$. We run synthetic data experiments on one Nvidia L40 GPU and each run consumes less than 2 minutes.

### A4.2 Regression Task Evaluation

In addition to the classification experiments, we evaluate our model on regression in this section.

#### A4.2.1 Implementation

**Data generation.** The regression target $y$ is generated from a uniform distribution $U(0, 4)$. We sample 4 latent invariant variables $\mathbf{c}$ from a normal distribution $N(y, \mathbf{I}_c)$. Two changing variables in the source domain $\mathbf{s}_{\text{src}}$ are sampled from a truncated Gaussian centered at the origin. In the target domain, changing variables $\mathbf{s}_{\text{tgt}}$ are sampled at multiple distances (e.g., $\{18, 24, 36\}$) from the origin. For dense shifts, observations $\mathbf{x}$ are generated by concatenating $\mathbf{c}$ and $\mathbf{s}$ and feeding them to a 4-layer MLP with ReLU activation. For sparse shifts, only two out of six dimensions of $\mathbf{x}$ are influenced by the changing variable $\mathbf{s}$. We generate 10k samples for training and 50 target samples for testing (one target sample accessed per run).

**Model.** We make two modifications on the classification model in Section 5. First, we substitute the classification head with a regression head (the last linear layer). Second, we replace the cross-entropy loss with MSE loss. We fix the loss weights of MSE loss and KL loss at 0.1 and 0.01 for all settings, respectively, and keep all other hyper-parameters the same as in the classification task. We use MSE as the evaluation metric.

#### A4.2.2 Results and Analysis

Table A1 displays the evaluation results. We can observe that our method consistently outperforms the baseline and maintains its performance over a wide range of shift distances. In contrast, the baseline that directly uses all the feature dimensions degrades drastically when the shift becomes severe. This indicates that our approach can indeed identify the invariant part of the latent representation, validating our theoretical results.

Table A2: **Hyperparameters for minimal change constraint in our experiments.**

| Dataset | $r$ | $ratio_l$ | $ratio_s$ |
|---------|-----|-----------|-----------|
| ImageNet | 4 | 5 | $1 \times 10^{-3}$ |
| CIFAR100 | 16 | 2 | $1 \times 10^{-1}$ |
| CIFAR10 | 64 | 1 | $1 \times 10^{-5}$ |

## A5 Real-world Data Experimental Details

### A5.1 Datasets

The datasets used in our experiments include CIFAR10-C, CIFAR100-C, ImageNet-C [45], and ImageNet100-C. CIFAR10-C and CIFAR100-C are extended versions of the CIFAR datasets [69] designed to evaluate model robustness against visual corruptions, featuring 10 and 100 classes respectively, each with 50,000 clean training samples and 10,000 corrupted test samples. ImageNet-C, on the other hand, scales this concept up with 1,000 classes, providing 50,000 test samples of each of 15 corruption types. ImageNet-100 [46] is a subset of ImageNet with 100 classes. In our experiments, we build ImageNet100-C by selecting 100 classes reported in Tian et al. [46] from ImageNet-C [45] with 15 different types of corruption.

### A5.2 Generative Adaptation with Entropy Minimization

When applying entropy minimization in the MAE-TTT framework [20], we did not directly integrate the entropy-minimization loss. The self-supervised MAE training process relies on masked images, whereas entropy minimization requires classifying the entire image. To address this, we introduced additional training steps using unmasked images and applied the entropy-minimization loss during these steps. Specifically, the training process for each test-time iteration is split into two stages: 1) Stage One: We follow the MAE-TTT approach by inputting masked images and training the model using reconstruction loss. In this stage, only the encoder is updated. 2) Stage Two: We input full images (32 in a batch) and optimize the model with the entropy minimization loss following SHOT [43]. In this stage, both the encoder and classifier are optimized. The learning rates for both stages are set to be the same. The experiments are conducted with the PyTorch 1.11.0 framework, CUDA 12.0 with 4 NVIDIA A100 GPUs.

### A5.3 Sparsity Regularization

Here, we provide the implementation details of our modification to add sparsity regularization. In the pre-train stage, we apply the ResNet50 [47] as the backbone network and follow [14, 44] to pre-train it on the clean CIFAR10, CIFAR100, and ImageNet training sets, with joint contrastive and classification losses. In the test-time adaptation process, we adopt the sequential TTA protocol as outlined in TTAC [44] and TeSLA [21]. This protocol prohibits the change of training objectives throughout the test phase. Moreover, all testing data be processed in a sequential manner (one-pass), ensuring each data point is passed through the adaptation process exactly once.

Our method is built upon TeSLA [21], and follows most of its hyperparameters. Thus, we only discuss the extra hyperparameters we involved, including the low-rank dimension $r$, the ratio of learning rate factor for soft frozen $ratio_l = \frac{\text{lr}_{lora}}{\text{lr}_{backbone}}$, and the ratio of sparsity loss $ratio_s$. The details are shown in Table A2. It was observed that the constraints on minimal change need to be more stringent as the complexity of the data increases. The experiments are conducted with the PyTorch 1.13.0 framework, CUDA 11.7 with an NVIDIA A100 GPU.

### A5.4 Recoverability of the Invariant Variable

We assess the impact of the recoverability of the invariant variable on Test-Time Adaptation (TTA) methods. To do this, we compare the performance of TTA methods using supervised and unsupervised pre-trained models that have similar ImageNet classification accuracy. Our goal is to validate whether invariant variables learned from annotated labels can improve test-time adaptation. We assume that

Table A3: **ImageNet-C evaluation of TTA algorithms with supervised and moco pre-trains .**

| TTA Algorithm | TENT [15] | EATA [61] | BN-adapt [71] | SAR [63] |
|---|---|---|---|---|
| Supervised pre-train | 38.13 | 42.77 | 29.13 | 41.47 |
| Moco pre-train | 8.94 | 1.25 | 20.13 | 12.23 |

Table A4: **Detailed corruption-wise results on CIFAR10-C, CIFAR100-C, and ImageNet-C**. We report the error rates (%) on 15 testing corruptions.

| Dataset | Method | Gaus. | Shot | Impu. | Defo. | Glas. | Moti. | Zoom | Snow | Fros. | Fog | Brig. | Cont. | Elas. | Pixe. | Jpeg | Avg. |
|---|---|---|---|---|---|---|---|---|---|---|---|---|---|---|---|---|---|
| CIFAR10-C | BN | 18.2 | 17.2 | 28.1 | 9.8 | 26.6 | 14.2 | 8.0 | 15.5 | 13.8 | 20.2 | 7.9 | 8.3 | 19.3 | 13.3 | 13.8 | 15.6 |
|  | Tent | 16.0 | 14.8 | 24.5 | 9.2 | 23.8 | 13.1 | 7.7 | 14.9 | 13.0 | 16.5 | 8.2 | 8.3 | 17.9 | 10.9 | 13.3 | 14.1 |
|  | SHOT | 16.5 | 15.3 | 23.6 | 9.0 | 23.4 | 12.7 | 7.5 | 14.0 | 12.4 | 16.1 | 7.5 | 8.0 | 17.4 | 12.5 | 13.1 | 13.9 |
|  | TTT++ | 18.0 | 17.1 | 30.8 | 10.4 | 29.9 | 13.0 | 9.9 | 14.8 | 14.1 | 15.8 | 7.0 | 7.8 | 19.3 | 12.7 | 16.4 | 15.8 |
|  | TTAC | 17.9 | 15.8 | 22.5 | 8.5 | 23.5 | 11.2 | 7.6 | 11.9 | 12.9 | 13.3 | 6.9 | 7.6 | 17.3 | 12.3 | 12.6 | 13.4 |
|  | TeSLA | 13.3 | 12.5 | 20.8 | 8.8 | 21.1 | 11.8 | 7.3 | 12.6 | 11.2 | 15.6 | 7.6 | 7.6 | 16.2 | 9.7 | 11.6 | 12.5 |
|  | TeSLA+MC | 13.0 | 12.6 | 19.8 | 8.8 | 19.9 | 10.9 | 7.5 | 12.2 | 11.0 | 14.4 | 7.2 | 7.4 | 15.4 | 9.2 | 10.9 | 12.0 |
| CIFAR100-C | BN | 48.2 | 46.4 | 61.1 | 33.8 | 58.2 | 41.4 | 31.9 | 46.1 | 42.5 | 54.7 | 31.3 | 33.3 | 48.4 | 39.0 | 39.6 | 43.7 |
|  | Tent | 43.3 | 41.2 | 52.7 | 31.2 | 50.8 | 36.1 | 29.3 | 41.9 | 38.9 | 43.6 | 30.1 | 31.0 | 43.5 | 34.4 | 36.5 | 39.0 |
|  | SHOT | 44.1 | 41.8 | 53.3 | 31.5 | 50.6 | 36.0 | 29.6 | 40.7 | 40.1 | 41.9 | 29.5 | 33.6 | 44.0 | 34.9 | 36.6 | 39.2 |
|  | TTT++ | 50.2 | 47.7 | 66.1 | 35.8 | 61.0 | 38.7 | 35.0 | 44.6 | 43.8 | 48.6 | 28.8 | 30.8 | 49.9 | 39.2 | 45.5 | 44.4 |
|  | TTAC | 47.7 | 45.7 | 58.1 | 32.5 | 55.3 | 36.6 | 31.2 | 40.3 | 40.8 | **44.7** | 30.0 | 39.9 | 47.1 | 37.8 | 38.3 | 41.7 |
|  | TeSLA | 40.0 | 38.9 | 51.5 | 32.2 | 49.1 | 36.9 | 29.7 | 40.4 | 37.4 | 46.0 | 29.3 | 30.7 | 42.7 | 32.9 | 34.6 | 38.2 |
|  | TeSLA+MC | 39.3 | 38.4 | 50.5 | 31.8 | 48.7 | 36.4 | 29.9 | 39.9 | 37.4 | 46.6 | 28.4 | 30.5 | 43.0 | 32.1 | 34.5 | 37.8 |
| ImageNet-C | BN | 83.5 | 82.6 | 82.9 | 84.4 | 84.2 | 73.1 | 60.5 | 65.1 | 66.3 | 51.5 | 34.0 | 82.6 | 55.3 | 50.3 | 58.7 | 67.7 |
|  | Tent | 70.8 | 68.7 | 69.1 | 72.5 | 73.3 | 59.3 | 50.8 | 53.0 | 59.1 | 42.7 | 32.6 | 74.5 | 45.5 | 41.6 | 47.8 | 57.4 |
|  | SHOT | 77.0 | 74.6 | 76.4 | 81.2 | 79.3 | 72.5 | 61.7 | 65.7 | 66.3 | 55.6 | 56.0 | 92.7 | 57.1 | 56.3 | 58.2 | 68.7 |
|  | TTAC | 71.5 | 67.7 | 70.3 | 81.2 | 77.3 | 64.0 | 54.4 | 51.1 | 56.9 | 45.4 | 32.6 | 79.1 | 46.0 | 43.7 | 48.6 | 59.3 |
|  | TTT++ | 69.4 | 66.0 | 69.7 | 84.2 | 81.7 | 65.2 | 53.2 | 49.3 | 56.2 | 44.4 | 32.8 | 75.7 | 43.9 | 41.6 | 46.9 | 58.7 |
|  | TeSLA | 65.0 | 62.9 | 63.5 | 69.4 | 69.2 | 55.4 | 49.5 | 49.1 | 56.6 | 41.8 | 33.7 | 77.9 | 43.3 | 40.4 | 46.6 | 55.0 |
|  | TeSLA+MC | 64.8 | 62.7 | 63.7 | 69.7 | 69.5 | 55.1 | 48.8 | 48.6 | 55.7 | 41.3 | 32.7 | 75.8 | 42.7 | 39.7 | 46.0 | 54.4 |

annotated labels can help learn better invariant variables $\mathbf{c}$, which play an important role in solving extrapolation problems.

For this detailed analysis, we employ ResNet50 models pre-trained in both supervised and self-supervised manners, such as MoCo [70]. For the MoCo model, we apply the linear probe to fit the classifier. For a fair comparison, we select checkpoints from both pre-training methods that have similar ImageNet accuracies, approximately 69.7%. We follow the open-source TTA Benchmark [71] to evaluate both models using different downstream TTA methods. ImageNet-C is used as the evaluation dataset, and all hyperparameters are set to their default values.

The performance of the different pre-trained models is summarized in Table A3. Methods using supervised pre-training outperform those using unsupervised pre-training by a significant margin, indicating that the invariant variables learned from annotated labels play a crucial role in enhancing test-time adaptation.

### A5.5 Additional quantitative results.

In Table A4, we provide the detailed performance with corruption-wise classification error rates on all CIFAR10-C, CIFAR100-C, and ImageNet-C datasets. Specifically, we report results under seed 0 on all 15 testing corruptions including Gaussian Noise, Shot Noise, Impulse Noise, Defocus Blur, Glass Blur, Motion Blur, Zoom Blur, Snow, Frost, Fog, Brightness, Contrast, Elastic Transformation, Pixelate, and JPEG Compression.

