# OpenReview forum: "Towards Understanding Extrapolation: a Causal Lens"
_NeurIPS.cc/2024/Conference — NeurIPS 2024 poster_

### Official Review · Reviewer_5kN3 · 2024-07-04

**Soundness:** 2
**Presentation:** 2
**Contribution:** 2
**Rating:** 4
**Confidence:** 2

**Summary:**

This work addresses the challenge of extrapolation in scenarios where only a few target samples are available. It aims to provide a theoretical understanding and methods for effective extrapolation without needing a target distribution within the training support. The approach involves a latent-variable model based on the minimal change principle in causal mechanisms. The study identifies conditions under which identification is possible even with a single off-support target sample. Experiments with synthetic and real-world data validate the theoretical and practical aspects of the findings.

**Strengths:**

1. The paper gives relatable motivating examples making it approachable for readers.
2. The paper is well-written, with a few minor typos.
3. The work tackles a very relevant problem in today's world.

**Weaknesses:**

1. Table 2 appears to omit results from TeSLA-s, which outperforms their TeSLA+SC. Thus their claims of SOTA are in question. Table 1 of:
     * Devavrat Tomar, Guillaume Vray, Behzad Bozorgtabar, and Jean-Philippe Thiran. Tesla: Test-time self-learning with automatic adversarial augmentation. In Proceedings of the IEEE/CVF Conference on Computer Vision and Pattern Recognition, pages 20341–20350, 2023.

2. The paper lacks expected sections. There is no related work, making it difficult to contextualize their work within the field. Additionally, some of the introduction could be better placed in a background section, specifically lines 41-67.

3. The authors do not provide any source code.

4. More elaboration on the limitations would enhance the paper. The authors seem to have overlooked reflecting sufficiently on potential limitations of their method. For instance, since everything operates in latent space, the method would only be applicable to models that possess a latent representation.

5. Minor typos:
     * What do the numbers in Table 1 represent? Accuracy? AUROC?
     * Line 127: Should it not be, "... values c as x_{src}"?
     * Line 127: Missing period.
     * Line 321: It would be helpful to provide a link to the code for MAE-TTT.
     * Line 356: Missing period.

**Questions:**

How would you apply your method for regression problems? Do you think it would perform well?

**Limitations:**

See weakness 4.

---

> ### Author Rebuttal · Authors · 2024-08-06
>
> Thank you for your valuable feedback and the time dedicated to reviewing our work. We address your concerns and questions as follows.
>
> >W1: “Table 2 appears to omit results from TeSLA-s.”
>
> Thank you for the comment. Thanks to your remark, we perform additional experiments to apply our sparsity constraint (SC) module to TeSLA-s and present the results as follows. We repeat our experiments over 3 random seeds and adopt the hyper-parameters from TeSLA+SC without any tuning.
>
> | Model   | CIFAR10-C     | CIFAR100-C    | ImageNet-C    |
> |---------|---------------|---------------|---------------|
> | Tesla-s | 12.1          | 37.3          | 53.1          |
> | Tesla-s+SC    | **11.7 ± 0.01** | **37.0 ± 0.06** | **50.9 ± 0.15** |
>
> We can observe that our module can consistently benefit existing methods (especially on the hardest dataset ImageNet-C), thus showcasing the applicability of our theoretical insights.
> We have included this result into Table 2 in our revised manuscript -- thank you for pointing us to this!
>
> We didn't include TeSLA-s in the initial draft because TeSLA-s relies on additional access to the source dataset during the adaptation (please also see this in Table 1 of Tomar et al.), which may not be as realistic and challenging as the source-free setting for TeSLA and TeSLA+SC.
>
> >W2: “The paper lacks expected sections.”
>
> Thank you for the constructive suggestion to help us improve the readability of our paper! In the submission, we deferred the related work section to Appendix A1. In light of your suggestion, we have condensed and placed it as Section 2 in our revision (original line 84 - 110) to aid contextualization (we reorganized spacing in Section 4 & 5 and moved Section 5.3 to the appendix to make space).
> Further, given your suggestion, we have condensed and merged formalisms (especially in lines 46-53 and 56-60) into the beginning of Section 2 in the revised version to increase the readability.
>
> >W3: source code.
>
> Thanks for the comment. We now provide source code for TeSLA+SC and have already passed them to the AC, following NeurIPS guidelines. Please let us know if you have any questions regarding the code, thank you!
>
> >W4: “More elaboration on the limitations.”
>
> Thank you for raising this concern. In light of your suggestion, we have expanded the limitation section in our revised paper. The added content is as follows.
>
> “On the theory aspect, the Jacobian norm utilized in Theorem 3.1 only considers the global smoothness of the generating function and thus may be too stringent if the function is much more well-behaved/smooth over the extrapolation region of concern. Therefore, one may consider a refined local condition to relax this condition.
> On the empirical side, our theoretical framework entails learning an explicit representation space. Existing methods without such a structure may still benefit from our framework but to a lesser extent. For instance, TeSLA (Table 2) receives a smaller boost compared to MAE-TTT (Table 3).
> Also, our framework involves several loss terms including reconstruction, classification, and the likelihood of the target invariant variable. A careful re-weighting over these terms may be needed during training.”
>
> >W5: "minor typos".
>
> Thank you so much for pointing them out! 1) Yes, it is classification accuracy -- we have added it to Table 1 caption. 2,3) We've reworded the sentence as "we need to identify the target sample $ \mathbf{x} _{\mathrm{tgt}} $ with source samples  $ \mathbf{x} _{\mathrm{src}} $ that share the same invariant variable values with the target sample, i.e., $ \mathbf{c} _{\mathrm{src}} = \mathbf{c} _{\mathrm{tgt}} $." 4,5) Thanks -- added both to the revision!
>
> >Q1: “regression problems.”
>
> Thank you for noting this. As our theory doesn’t place specific assumptions on the conditional distribution $ p( y | \mathbf{c} ) $, the framework is rather flexible to accommodate regression problems.
>
> Specifically, the procedure to learn the disentangled invariant variable $ \hat{\mathbf{c}} $ through objectives (3) or (4) is totally unsupervised and thus agnostic to $y$’s distribution. After learning $ \hat{\mathbf{c}}$, we can choose to train a regressor on pairs $ (\hat{\mathbf{c}}, y) $ from the source distribution.
>
> Thanks to your question, we have included the following synthetic experiments on regression.
>
> **Data Generation:** The regression target $y$ is generated from a uniform distribution $U(0,4)$. We sample 4 latent invariant variables $\mathbf{c}$ from a normal distribution $N(y, I_c)$. Two changing variables in the source domain $\mathbf{s} _{\mathrm{src}}$ are sampled from a truncated Gaussian centered at the origin. In the target domain, changing variables $\mathbf{s} _{\mathrm{tgt}}$ are sampled at multiple distances (e.g., $\{18, 24, 36\}$) from the origin. Observations $\mathbf{x}$ are generated by concatenating $\mathbf{c}$ and $\mathbf{s}$ and feeding them to a 4-layer MLP with ReLU activation. We generate 10k samples for training and 50 target samples for testing (one target sample accessed per run).
>
> **Model:** We make two modifications on the classification model in the paper. First, we change the classification head to a regression head (the last linear layer). Second, we replace the cross-entropy loss with MSE loss. We fix the loss weights of MSE loss and KL loss at 0.1 and 0.01 for all settings, respectively, and keep all other hyper-parameters the same as in the classification task. We use MSE as the evaluation metric.
>
> **Results:** The results are summarized below, which indicate that the proposed method can be extended to the regression setting.
>
> | Dense Shift Distance | 18       | 24       | 30       |
> |---------|----------|----------|----------|
> | Baseline| 1.6400   | 2.4430   | 3.2627   |
> | Our Method | 1.4006   | 1.6039   |1.6812   |
>
> We are also working on more settings and will update you once the results are available.
>
> ---
> Please let us know if you have further questions -- thank you so much!

---

> > ### Author Response · Authors · 2024-08-08
> > **More results on the regression setting**
> >
> > We now provide more results on the regression task we’ve included in the rebuttal. This setup corresponds to the sparse shift setting in our theory where only two out of six dimensions of $\mathbf{x}$ are influenced by the changing variable $\mathbf{s}$. In comparison, the setting we’ve included in the rebuttal corresponds to the dense shift case. The distinction is described in lines (lines 300-304). The other setups are identical. We directly adopt the hyper-parameters from the dense shift setting without modifications. The results are also measured in MSE (the lower the better).
> >
> > | Sparse Shift Distance | 18     | 24     | 30     |
> > |----------------------|--------|--------|--------|
> > | Baseline             | 1.835 | 3.323 | 5.841 |
> > | Our Method           | 1.145 | 1.476 | 1.596 |
> >
> > We can observe that like the dense shift case, our method outperforms the baseline consistently and maintains the performance over a wide range of shift severities. In contrast, the baseline that directly uses all the feature dimensions degrades drastically when the shift becomes severe. This indicates that our approach can indeed identify the invariant part of the latent representation, validating our theoretical results.
> >
> > ---
> >
> > Please let us know if we have resolved your concerns – thank you!

---

> > > ### Author Response · Authors · 2024-08-12
> > >
> > > Dear Reviewer 5kN3,
> > >
> > > As the discussion deadline approaches, we are wondering whether our responses have properly addressed your concerns? Your feedback would be extremely helpful to us. If you have further comments or questions, we hope for the opportunity to respond to them.
> > >
> > > Many thanks,
> > >
> > > 7667 Authors

---

> > > > ### Author Response · Authors · 2024-08-13
> > > >
> > > > ​​Once again, we are grateful for your time and effort for reviewing our paper. Since the discussion period will end in around a day, we are very eager to get your feedback on our response. We understand that you are very busy, but we would highly appreciate it if you could take into account our response when updating the rating and having a discussion with AC and other reviewers.
> > > >
> > > > Thanks for your time,
> > > >
> > > > Authors of # 7667

---

> > > > > ### Author Response · Authors · 2024-08-14
> > > > >
> > > > > Dear reviewer 5kN3,
> > > > >
> > > > > Since the discussion period will end in a few hours, we will be online waiting for your feedback on our rebuttal, which we believe has fully addressed your concerns.
> > > > >
> > > > > We would highly appreciate it if you could take into account our response when updating the rating and having discussions with AC and other reviewers.
> > > > >
> > > > > Thank you so much for your time and efforts. Sorry for our repetitive messages, but we're eager to ensure everything is addressed.
> > > > >
> > > > > Authors of # 7667

---

### Official Review · Reviewer_gbwX · 2024-07-11

**Soundness:** 2
**Presentation:** 3
**Contribution:** 3
**Rating:** 6
**Confidence:** 3

**Summary:**

The paper addresses the problem of out-of-support extrapolation and presents identification results within a latent variable framework. This framework resembles invariant learning, where one subset of latent variables directly causes the labels, while another subset, termed style latents, undergoes a distribution shift at test time. The authors provide identification results for scenarios where this distribution shift affects the entire image (global shift case) and where it affects only specific regions of the image (local shift case). Additionally, they connect their identification results to the entropy maximization objective used in test time adaptation (TTA) algorithms. These theoretical aspects are verified through experiments on popular TTA benchmarks, such as CIFAR-C and ImageNet-C.

**Strengths:**

* The identification results for out-of-support extrapolation presented in the paper appear to be novel, to the best of my knowledge.

* The paper is overall well-written, with clearly explained assumptions required for the identification results, and detailed descriptions of the experimental setup and results.

**Weaknesses:**

* My main concern is with the technical soundness of the paper, particularly regarding certain claims and connections. I believe the connection between the identification results and the test-time adaptation (TTA) objective is problematic.  Specifically, Theorems 3.2 and 3.4 require that the learned distribution of the data ($\hat{p}(x)$) matches the true distribution ($p(x)$). However, the TTA methods are not generative models, and therefore they do not estimate the distribution of the data. However, TTA methods are not generative models and do not estimate data distributions, so they cannot enforce the constraint $p(x) = \hat{p}(x)$.  Therefore, it is unclear how the identification results provide insights into TTA methods. Identification concerns the existence of a unique solution for perfect estimation under the learning objective, but the learning objectives for the identification analysis and TTA methods are different. While the authors discuss this in Section 3.3, however, the issue is deeper,  they do not address the density estimation constraint and its connection to test-time adaptation comprehensively.

* The definition of identification considered by the authors (Definition 2.1) is problematic because it does not relate the estimated latents ($\hat{c}$) to the true latents ($c$). This implies that the estimated latents could be an arbitrary function of the true latents, which is concerning as it makes it unclear why such identifiability would be desirable. The identification criteria state that the estimated content latents are the same whenever the true content latent variables are the same. However, this does not necessarily imply a meaningful relationship between the estimated and true latents. For instance, block identification [1] ensures that the estimated content latents are a function of only the true content latents, establishing a clear relationship. If we cannot relate the estimated latents to the true latents, it is unclear whether the inferred content latents ($\hat{c}$) are genuinely informative about the true content latents ($c$). This lack of a meaningful connection undermines the practical value of the identifiability results.

References:

[1] Lachapelle, Sébastien, Divyat Mahajan, Ioannis Mitliagkas, and Simon Lacoste-Julien. "Additive decoders for latent variables identification and cartesian-product extrapolation." Advances in Neural Information Processing Systems 36 (2024).

[2] Khemakhem, Ilyes, Diederik Kingma, Ricardo Monti, and Aapo Hyvarinen. "Variational autoencoders and nonlinear ica: A unifying framework." In International conference on artificial intelligence and statistics, pp. 2207-2217. PMLR, 2020.

[3] Zimmermann, Roland S., Yash Sharma, Steffen Schneider, Matthias Bethge, and Wieland Brendel. "Contrastive learning inverts the data generating process." In International Conference on Machine Learning, pp. 12979-12990. PMLR, 2021.

**Questions:**

Please refer to the weaknesses section in my review above for more details regarding my concerns and queries.

* Why does the identification analysis apply for TTA when its learning objective is different from the learning objective considered in the identification analysis?

* How do the identification criteria considered in this study relate to standard identification criteria used in causal representation learning/ non-linear ICA [1, 2, 3]? Does satisfying the proposed criteria imply we learn meaningful content variables such that they are related to the true content variables?

* Following the results in Table 2 regarding the effect of adding sparsity regularization, it is unclear how TeSLA + SC provides an improvement over TeSLA, as the difference in mean performance between the two approaches is within the standard error. The authors should consider revising line 357 to accurately reflect this observation.

**Limitations:**

Yes, the authors have properly addressed the limitations and impact of their work.

---

> ### Author Rebuttal · Authors · 2024-08-06
>
> Thank you for your detailed review and the valuable time you have dedicated to our work.
>
> It seems that the reviewer might have seen a previous version of this manuscript. In that case, please let us kindly highlight that this submission is significantly different: in the earlier instance, we tried justifying TTA with our theory, which we later realized was not optimal.
>
> In this submission, the main contribution is to provide a fundamental understanding of extrapolation and discuss principled ways to address it, independent of existing TTA algorithms. To support it, we substantially rewrote the introduction, problem formulation, and analysis of the identification theory. Further, we implemented our framework with synthetic experiments in Sec 4 to rigorously validate our theory. Additionally, for real-world experiments, we base our implementation on the autoencoder-based MAE-TTT, whose reconstruction loss facilitates matching the estimated distribution $\hat{p}(\mathbf{x})$ and the true marginal distribution match $p(\mathbf{x})$ in our objective.
>
> Thus, we feel the review comments mainly apply to the earlier version, not this one. We apologize for any confusion caused by the multi-version problem may have caused and appreciate your further feedback.
>
> >W1& Q1: “...the connection between the identification results and the test-time adaptation (TTA) objective is problematic..”
>
> Thank you for raising this concern. Please kindly note that we do not claim to justify or explain existing TTA methods with our identification theory. Instead, we discuss the relationship between our theoretical framework on extrapolation and TTA methods to identify gaps (lines 281-286) and explore how our ideas may benefit them. Our experiments show that incorporating our insights (aligning the invariant variable in Sec 5.1 and sparsity in Sec 5.2) can improve TTA methods.
>
> In Sec 3.2 and the main experiments in Sec 5.1, we focus on the autoencoder-based model MAE-TTT, where the reconstruction loss facilitates matching the estimated distribution $\hat{p}(\mathbf{x})$ with the true marginal distribution $p(\mathbf{x})$, especially in the extreme case where the reconstruction is perfect.
> Even with our modifications, TTA algorithms may not fully adhere to the theoretical framework. Nonetheless, we hope this serves as a meaningful step towards developing more principled extrapolation methods for real-world datasets.
>
> >W2 & Q2: “The definition of identification considered by the authors (Definition 2.1) is problematic.”
>
> Thank you for the question. Our identification notion (Definition 2.1) implies the block-wise identifiability [1] – there exists an invertible map from the true invariant variable $\mathbf{c}$ to the estimated invariant variable $\hat{\mathbf{c}}$. In light of your feedback, we’ve added the following text to line 120 to make it clearer: “which is equivalent to the blockwise identifiability in prior work [1].”
>
> We give a proof that Definition 2.1 implies block-wise identifiability.
>
> As both the generating function $g$ and the estimated one $\hat{g}$ are invertible (Assumption 3.1), we have an invertible map $ h: ( \mathbf{c}, \mathbf{s} ) \mapsto ( \hat{\mathbf{c}}, \hat{\mathbf{s}} ) $.
>
> We first show that Definition 2.1 implies that $\hat{ \mathbf{c}}$ depends only on $ \mathbf{c}$. Suppose, for contradiction, that $\hat{ \mathbf{c}}$ depends on both $ \mathbf{c}$ and $ \mathbf{s}$.
> Then there would exist $  \mathbf{c} _{o}$, $ \mathbf{s} _{1}$, and $ \mathbf{s} _{2}$ $( \mathbf{s} _{1} \neq  \mathbf{s} _{2}$) such that: $h( \mathbf{c} _{o},  \mathbf{s} _{1}) = (\hat{ \mathbf{c}} _{1}, \hat{ \mathbf{s}} _{1})$ and $h( \mathbf{c} _{o},  \mathbf{s} _{2}) = (\hat{ \mathbf{c}} _{2}, \hat{ \mathbf{s}} _{2})$ where $\hat{ \mathbf{c}} _{1} \neq \hat{ \mathbf{c}} _{2}$.
>
> However, this contradicts Definition 2.1, because we have $ \mathbf{c} _{o} =  \mathbf{c} _{o} $, but $\hat{\mathbf{c}} _{1} \neq \hat{\mathbf{c}} _{2}$. Therefore, the proposition must be false, and $\hat{ \mathbf{c}}$ must depend only on $ \mathbf{c}$.
>
> Now we define $h_{c}:  \mathbf{c} \mapsto \hat{ \mathbf{c}}$ as follows: for any $ \mathbf{c}$, choose any $ \mathbf{s}$ and let $h_{c}( \mathbf{c}) = \hat{ \mathbf{c}}$, where $(\hat{ \mathbf{c}}, \hat{ \mathbf{s}}) = h( \mathbf{c},  \mathbf{s})$. This is well-defined because we've proven that $\hat{ \mathbf{c}}$ only depends on $ \mathbf{c}$, so the choice of $ \mathbf{s}$ doesn't matter.
>
> We finally show that $h_{c}$ is invertible by noting the following.
>
> $h_{c}$ is injective: Let $ \mathbf{c} _{1},  \mathbf{c} _{2}$ be in the domain of $h _{c}$. If $h _{c}( \mathbf{c} _{1}) = h _{c}( \mathbf{c} _{2})$, then $\hat{ \mathbf{c}} _{1} = \hat{ \mathbf{c}} _{2}$ by definition.  By Definition 2.1, this implies $ \mathbf{c} _{1} =  \mathbf{c} _{2} $. Therefore, $h _{c}$ is injective.
>
> $h_{c}$ is surjective: Let $\hat{ \mathbf{c}}$ be in the image of $h_{c}$. Since $h$ is invertible, there exists $( \mathbf{c},  \mathbf{s})$ such that $h( \mathbf{c},  \mathbf{s}) = (\hat{ \mathbf{c}}, \hat{ \mathbf{s}})$ for some $\hat{ \mathbf{s}}$. By definition of $h_{c}$, $h_{c}( \mathbf{c}) = \hat{ \mathbf{c}}$. Therefore, $h_{c}$ is subjective.
>
> Please let us know if we have fully addressed your concerns and we’d be delighted to discuss further.
>
> >Q3: “it is unclear how TeSLA + SC provides an improvement over TeSLA, as the difference in mean performance between the two approaches is within the standard error.”
>
> Thanks for the comment. We were wondering if there was a misreading – the improvements of TeSLA+SC over TeSLA are $0.4$. $0.2$, and $0.5$, all larger than the std $0.1$.
> As we acknowledge in lines 357-359, these improvements, though modest, are consistent and thus demonstrate the applicability of our approach beyond autoencoding TTA approaches like MAE-TTT.
>
> ---
>
> We are eager to hear your feedback. We’d deeply appreciate it if you could let us know whether your concerns have been addressed.

---

> ### Comment · Reviewer_gbwX · 2024-08-11
>
> Thanks for your detailed response! I have a better understanding of the implications of the theoretical results and how authors intended to connect them with prior learning methods for TTA (hence adjusted my rating as well). A suggestion would be to change the introduction and especially the contribution to highlight this more, it is heavily centered around the connection between theoretical results and the existing TTA approaches. The authors could mention more explicitly that they propose an improvement over MAE-TTT inspired by their theory and evaluate it on TTA benchmarks. Furthermore, section 3.3 can be improved and perhaps made into a separate section, with more details MAE-TTT being related to the theoretical analysis and proposed changes on top of it.
>
> I would be most interested in understanding the performance of MAE-TTT +  since it directly conforms to the theory. This prompts another question: why did the authors not experiment with the proposed MAE-TTT + entropy minimization for benchmarks in Table 2? MAE-TTT is only explored in section 5.1 on the ImageNet100-C dataset and is not included for the benchmarks in Section 5.2 Wouldn't that be an important question to understand how the approach following the theoretical analysis compares with other state-of-the-art approaches? I am happy to increase my score further if the authors can perform experiments for the same.

---

> ### Comment · Reviewer_gbwX · 2024-08-11
>
> Also, thanks for the proof on Definition 2 and its connection with block identifiability! The argument is correct and addresses my concerns.
>
> Regarding my final question about improvement with sparsity regularization over TeSLA, thanks for the clarification, my earlier statement was incorrect. However, I am still not convinced since it also depends on the standard deviation in the performance of the method TesLA as well. For example, if the performance of TeSLA on CIFAR10-C is $12.5 \pm 0.3$, then the confidence intervals would overlap. But I do not see this as a major concern though I encourage authors to report the deviation in the error rate of all baselines as well.

---

> ### Author Response · Authors · 2024-08-12
>
> Thank you for your detailed feedback and we are delighted that we have cleared your previous concerns!
>
> > Writing suggestions.
>
> We highly appreciate your suggestion on the writing. Thanks to your suggestion, we have made the following modifications:
>
> 1. We have replaced current lines 70-72 with: “In particular, we apply our theoretical insights to improve autoencoder-based MAE-TTT [24] and observe noticeable improvements on TTA tasks. We also demonstrate that basic principles (sparsity constraints) from our framework can benefit state-of-the-art TTA approach TeSLA [27].”
>
> 2. We have replaced the contribution lines 82-83 with: “Inspired by our theory, we propose to add a likelihood maximization term to autoencoder-based MAE-TTT [24] to facilitate the alignment between the target sample and the source distribution. In addition, we propose sparsity constraints to enhance state-of-the-art TTA approach TeSLA [27]. We validate our proposals with empirical evidence.”
>
> 3. We have re-organized lines 275-287 in Section 3.3 to make the relationship with MAE-TTT clearer. In particular, we make lines 276-280 as a separate paragraph to highlight the connection (the reconstruction objective) and lines 280-287 another paragraph to specify the distinction and our proposal, starting with “Despite the resemblance on the reconstruction objective, MAE-TTT doesn’t explicitly perform the representation alignment as our objectives (2)(3).”
>
> We hope these modifications would make the message clearer and we would appreciate your further feedback!
>
>
> > MAE-TTT +entropy for other benchmark datasets.
>
> Thanks for the great question!
>
> In light of your question, we have started running MAE-TTT + entropy minimization on ImageNet-C. Given our computing resource, MAE-TTT on ImageNet will require 20 days to complete and we may not be able to provide full results by the end of the discussion period. Nevertheless, we will include this result in our revision. Thank you for your understanding!
>
> For CIFAR-10/100-C: MAE-TTT requires pre-trained and fine-tuned MAE checkpoints, which are only available for ImageNet (please see the original paper [a]). We are concerned that directly using the ImageNet pre-trained checkpoints on CIFAR-10/100-C would lead to unfair comparisons and also require a significant amount of fine-tuning for the transfer between datasets. Thus, we decide to focus on ImageNet-C dataset for now, and this is the most representative and challenging benchmark.
>
> We thank you for your understanding and patience!
>
> [a] Masked Autoencoders Are Scalable Vision Learners. He et al. CVPR 2022.

---

> > ### Author Response · Authors · 2024-08-12
> >
> > > Proof on Definition 2.
> >
> > Thank you for your positive feedback and we are happy to see this major concern resolved!
> >
> > > Standard deviations for TeSLA.
> >
> > Thanks for the question! We’ve included in our revision standard deviations for TeSLA performances as follows.
> > | Method | CIFAR10-C   | CIFAR100-C  | ImageNet-C |
> > |--------|-------------|-------------|------------|
> > | TeSLA  | 12.5 ± 0.04 | 38.2 ± 0.03 | 55.0 ± 0.17 |
> > Just as you can see, the standard deviations are all relatively small, so they should not affect our conclusion.
> >
> > ---
> >
> > Thank you for your thoughtful comments and remarks and we’d highly appreciate your further feedback!

---

### Official Review · Reviewer_HS3a · 2024-07-12

**Soundness:** 3
**Presentation:** 3
**Contribution:** 3
**Rating:** 7
**Confidence:** 3

**Summary:**

The authors discuss the problem of domain adaptation or distribution shift in the case when only a single point in the shifted domain is available rather than the full distribution. The authors approach such a problem from the perspective of a latent variable model, which assumes that the observations are generated from two latent variables: one is invariant under domain shifts (and carries informations about the classification label) and the other one changes under domain shifts but is irrelevant for classification. However, the generator function mixes these two latent variables making the classification under domain shifts difficult. The authors theoretically discuss the identifiability of the invariant latent variables in two scenarios, when the irrelevant for classification latent variables affects all dimensions in the input or only a subset of them. These theoretical results are validated in experiments on synthetic and real-world data.

**Strengths:**

+ Interesting and practically relevant problem
+ The paper is clearly written and easy to follow
+ Theoretical results formalising the intuition of invariant and nuisance latent variables

**Weaknesses:**

- I am not sure I completely understood the connection between the theory and the actual learning objective. Specifically, I would appreciate if the authors could elaborate a bit more on how the Eq. (2) and (3) are connected to the learning objective in Sec. 3.3 (also see the Questions part of the review)

**Questions:**

- It is very interesting that you derive a bound of how far the target point can be from the source manifold in Assumption 3.1-v. I wonder how tight do you think this bound is?
- Could you provide a bit more intuition why Assumption 3.3.-iv allows us to "identify the unaffected dimension indices [...] with our estimated model", I am not sure I understood this point?
- Did I understand correctly that in practice the estimation algorithms (as described in Sec. 3.3.) reduce to training an auto-encoder to match the data distribution, and a classifier in the latent space? So the key idea is that "raising" the classification to the latent space allows us to separate invariant latents from the nuisance variables?

**Limitations:**

The limitations are adequately addressed

---

> ### Author Rebuttal · Authors · 2024-08-06
>
> Thank you for your thoughtful review and valuable questions! We address your questions point-to-point in the following.
>
> > W1 & Q3: “... how the Eq. (2) and (3) are connected to the learning objective in Sec. 3.3”, “Did I understand correctly that in practice the estimation algorithms (as described in Sec. 3.3.) reduce to training an auto-encoder to match the data distribution, and a classifier in the latent space? So the key idea is that "raising" the classification to the latent space allows us to separate invariant latents from the nuisance variables?”
>
> Thank you for the great question! You are absolutely right about the general implementation framework. The matching distribution constraint in (2) (3) entails learning a generative model with a representation space $\hat{\mathbf{z}}$ and maximizing likelihood of the target invariants $\hat{p} ( \hat{\mathbf{c}}_{\mathrm{tgt}} )$ enables us to separate the invariant latents $\mathbf{c}$ from the nuisance variables $\mathbf{s}$, as you pointed out precisely. Given this disentangled representation, we can train a classifier on the invariant latents from the source distribution, which can be directly applied to the target distribution. Please let us know if you’d like elaboration – thanks!
>
>
>
>
> > Q1: “It is very interesting that you derive a bound of how far the target point can be from the source manifold in Assumption 3.1-v. I wonder how tight do you think this bound is?”
>
> We appreciate your insightful question. This bound is tight in a sense that the equality can be attained in worst-case scenarios. This is because beyond the threshold in the bound, a manifold characterized by a distinct invariant variable  $ \mathbf{c} \neq \mathbf{c} _{\mathrm{tgt}} $ may also explain the target sample $\mathbf{x} _{\mathrm{tgt}}$. This ambiguity would thwart our attempt to uniquely determine the invariant latent of $ \mathbf{x} _{\mathrm{tgt}} $.
> To aid intuition, let’s look at Figure 1b. The threshold in the bound exactly characterizes the starting point of the “unidentifiable region”. In the worst case when the target sample $\mathbf{x} _{\mathrm{tgt}}$ lands in the doubly shaded area, we cannot uniquely identify which manifold it belongs to without further assumptions/knowledge.
>
>
> > Q2: “Could you provide a bit more intuition why Assumption 3.3.-iv allows us to "identify the unaffected dimension indices [...] with our estimated model", I am not sure I understood this point?”
>
> Thank you for the question! Assumption 3.3 iv enforces that the unaffected pixels exhibit certain dependence among them, in the sense that if we divide this region into any two partitions and generate these two regions separately, we would need more “capacity” (i.e., Jacobian ranks) than generating this region jointly, because in the latter case some information would be shared between the partitions. Further note that Assumption 3.3 iii coerces the affected region to be either small or disjoint from the unaffected region. Thus, the cross-region dependence (between the unaffected and the affected regions) is small. Intuitively, this clear contrast offers us the signal to disentangle these two regions.
>
> For instance, in Figure 1c, the pixels within the region of the cow are highly dependent, whereas the cow region and the background region are significantly less so. Thus, this contrast aids our humans to distinguish the two regions rather easily.
>
> ---
> Please let us know if you’d like further illustration, thank you!

---

> > ### Comment · Reviewer_HS3a · 2024-08-13
> >
> > Thank you very much for a thorough rebuttal! I confirm my positive view on this paper and happy to increase my score.

---

> > > ### Author Response · Authors · 2024-08-13
> > >
> > > Thank you so much for your encouraging words and we are truly grateful for your dedicated time and constructive comments!

---

### Official Review · Reviewer_zpbG · 2024-07-13

**Soundness:** 3
**Presentation:** 2
**Contribution:** 4
**Rating:** 6
**Confidence:** 3

**Summary:**

* With regard to extrapolability in classification problems, they introduce a reasonable assumption that the latent factor generating the distribution shift only affects the input x but does not affect the label y is introduced and the identifiability of the latent factor is proved under additional reasonable assumptions.
* The interaction between the smoothness of the generating function, the distance out of support, and the nature of the shift (is the shift restricted to some pixels of the image?) is clarified.
* They discuss the relationship with test-time adaptation, extend using sparsity constraint, and verify its empirical performance.

**Strengths:**

* The relation between extrapolation and causality is a hot research topic.
* The motivation behind the assumption is clear that the class label is related only to the inviant factors: "factors such as camera angles and lighting do not affect the object’s class in an image."
* Under the reasonable assumption, an interesting theoretical result on the identifiability of the latent invariant factors is shown.
* The existing test-time adaptation is shown to be related to this theory and extended based on the implication; the sparsity is added in the adaptation, which shows superior empirical performance.

**Weaknesses:**

* Methodological improvement and its empirical superiority to the existing test-time adaptation itself is marginal.

**Questions:**

Q1. Is the reported error bar in Table 2 standard error or standard deviation?
Q2. Does it matter if the marginal distribution of the label $p(y)$ changes in the test phase?

**Limitations:**

Most of the important results are for the image classification problem, where the class-conditioned distribution p(x|y) is separated for each y and x has much information, and thus only unchanged pixels have enough information of $y$ under the sparse change assumption. It seems not to be simply extended to numerical predictions.

---

> ### Author Rebuttal · Authors · 2024-08-06
>
> Thank you for your encouraging words and valuable feedback! Below, we address your questions and indicate the changes we’ve made thanks to your suggestion.
>
> > W1: “Methodological improvement and its empirical superiority to the existing test-time adaptation itself is marginal.”
>
> Thank you for your feedback. Please kindly note that our main empirical results (Table 3) demonstrate rather significant performance gain over the baseline ($+4.87 \\%$). This is the case where the base method (MAE-TTT) is a representation-based approach and thus conforms to our theoretical model (please see Section 3.3), thus directly substantiating our theoretical insights.
>
> Table 2 shows that our theoretical insights can consistently benefit a broader class of TTA approaches, even if they don’t exactly conform to our framework. Although some gains are marginal, we believe that these results demonstrate the generality of our theoretical insight.
>
> >Q1:  “Is the reported error bar in Table 2 standard error or standard deviation?”
>
> It is the standard deviation over three random seeds. Thanks to your reminder, we’ve already included this in the figure caption in our revised manuscript.
>
> >Q2. “Does it matter if the marginal distribution of the label $p(y)$ changes in the test phase?”
>
> Thank you for the interesting question. The marginal shift of $p(y)$ would not affect the model performance, under proper assumptions of the shift (please note assumptions are generally necessary to avoid arbitrary shifts).
>
> To be specific, under our graphical formulation in which $y$ is a child of $\mathbf{c}$,  the shift of $p(y)$ corresponds to that of the support-invariant variable $p(\mathbf{c})$. If the shift doesn’t change the support of $p(\mathbf{c})$ but only its density on the shared support, our guarantee will still hold true with sufficient samples. This is because this conditional distribution $ p( y | \mathbf{c} )$ can still be learned and transferable as in our case.
>
> > Q3: “Most of the important results are for the image classification problem…It seems not to be simply extended to numerical predictions.”
>
> Thank you for noting this. As our theory doesn’t place specific assumptions on the conditional distribution $ p( y | \mathbf{c} ) $, the framework is rather flexible to accommodate regression problems.
>
> Specifically, the procedure to learn the disentangled invariant variable $ \hat{\mathbf{c}} $ through objectives (3) or (4) is totally unsupervised and thus agnostic to $y$’s distribution. After learning $ \hat{\mathbf{c}}$, we can choose to train a regressor on pairs $ (\hat{\mathbf{c}}, y) $ from the source distribution.
>
> Thanks to your question, we have included in our manuscript the following synthetic experiments on regression tasks.
>
> **Data Generation:** The regression target $y$ is generated from a uniform distribution $U(0,4)$. We sample 4 latent invariant variables $\mathbf{c}$ from a normal distribution $N(y, I_c)$. Two changing variables in the source domain $\mathbf{s} _{\mathrm{src}}$ are sampled from a truncated Gaussian centered at the origin. In the target domain, changing variables $\mathbf{s} _{\mathrm{tgt}}$ are sampled at multiple distances (e.g., $\{18, 24, 36\}$) from the origin. Observations $\mathbf{x}$ are generated by concatenating $\mathbf{c}$ and $\mathbf{s}$ and feeding them to a 4-layer MLP with ReLU activation. We generate 10k samples for training and 50 target samples for testing (one target sample accessed per run).
>
> **Model:** We make two modifications on the classification model in the paper. First, we change the classification head to a regression head (the last linear layer). Second, we replace the cross-entropy loss with MSE loss. We fix the loss weights of MSE loss and KL loss at 0.1 and 0.01 for all settings, respectively, and keep all other hyper-parameters the same as in the classification task. We use MSE as the evaluation metric.
>
>
> **Results:** The results are summarized below, which indicate that the proposed method can be extended to the regression setting.
>
> | Dense Shift Distance | 18       | 24       | 30       |
> |---------|----------|----------|----------|
> | Baseline| 1.6400   | 2.4430   | 3.2627   |
> | Our Method | 1.4006   | 1.6039   |1.6812   |
>
>
> We are also working on more settings and will update you once the results are available.
>
> ---
>
> Please let us know if we have properly addressed your questions and we are more than happy to discuss more!

---

> > ### Author Response · Authors · 2024-08-08
> > **More results on the regression setting**
> >
> > We now provide more results on the regression task we’ve included in the rebuttal. This setup corresponds to the sparse shift setting in our theory where only two out of six dimensions of $\mathbf{x}$ are influenced by the changing variable $\mathbf{s}$. In comparison, the setting we’ve included in the rebuttal corresponds to the dense shift case. The distinction is described in lines (lines 300-304). The other setups are identical. We directly adopt the hyper-parameters from the dense shift setting without modifications. The results are also measured in MSE (the lower the better).
> >
> > | Sparse Shift Distance | 18     | 24     | 30     |
> > |----------------------|--------|--------|--------|
> > | Baseline             | 1.835 | 3.323 | 5.841 |
> > | Our Method           | 1.145 | 1.476 | 1.596 |
> >
> > We can observe that like the dense shift case, our method outperforms the baseline consistently and maintains the performance over a wide range of shift severities. In contrast, the baseline that directly uses all the feature dimensions degrades drastically when the shift becomes severe. This indicates that our approach can indeed identify the invariant part of the latent representation, validating our theoretical results.
> >
> > ---
> >
> > Please let us know if we have resolved your concerns – thank you!

---

> > > ### Author Response · Authors · 2024-08-13
> > >
> > > Dear Reviewer zpbG,
> > >
> > > We were wondering whether your technical had been properly addressed by our responses so far. Please let us know if you have further questions or concerns that we can address. Thank you for your engagement with our work.
> > >
> > > Many thanks,
> > >
> > > Authors of 7667

---

> > > > ### Comment · Reviewer_zpbG · 2024-08-14
> > > >
> > > > Thank you for your response. My concerns have been addressed. I have already scored high enough and will maintain my score.

---

> ### Author Response · Authors · 2024-08-14
> **Thanks a lot for your recognition and valuable suggestions**
>
> Dear Reviewer zpbG,
>
> We really appreciate your recognition of our work and your kind words, and we are happy to hear that your concerns have been addressed. Again, thank you for your valuable suggestions which have undoubtedly contributed to improving the quality of our paper.
>
> Many thanks,
>
> The authors of #7667

---

### Author Rebuttal · Authors · 2024-08-07

We are grateful to all reviewers for their efforts and helpful comments regarding our paper. We are encouraged that Reviewers **zpbG**, **HS3a**, and **5kN3** find our problem setup relevant and interesting, Reviewers **zpbG** and **gbwX** find our theory interesting and novel, and all Reviewers find our paper clearly written.


Below is a summary of our responses:

* To **Reviewer zpbG**: We have provided further details and introduced a new experiment focusing on the regression task.
* To **Reviewer HS3a**: We have detailed the connections between our theoretical framework and the practical learning objectives and assumptions.
* To **Reviewer gbwX**: We have clarified the key contributions of our formulation and theory.
* To **Reviewer 5kN3**: We have included new experiments using TeSLA-s as a baseline and in a regression setting. We have also shared the source code for TeSLA+SC and TeSLA-s+SC, which has been forwarded to ACs. Further, we have expanded the limitation section, moved the related works from the Appendix to Section 2, and corrected the typos.


Please review our detailed responses to each point raised. We hope that our revisions and clarifications satisfactorily address the concerns raised. We thank you again for your valuable time and expertise.

---

### Decision · Program_Chairs · 2024-09-25

**Decision:**

Accept (poster)

**Comment:**

After reviewing the paper, the feedback from the reviewers, and the authors' thorough rebuttals, I recommend accepting the paper. The authors have effectively addressed the concerns raised, including those from Reviewer 5kN3 (I read both the review and rebuttal). The work tackles an important problem and presents a solid theoretical foundation with empirical validation. Given these strengths, I believe the paper merits acceptance.